



# An urban microwave link rainfall measurement campaign

Thomas C. van Leth[1], Aart Overeem[1,2], Remko Uijlenhoet[1], Hidde Leijnse[2]

[1]Hydrology and Quantitative Water Management Group, Wageningen University, P.O. Box 47, 6700 AA, Wageningen, the Netherlands

[2]Royal Netherlands Meteorological Institute (KNMI), P.O. Box 201, 3730 AE, De Bilt, the Netherlands

*Correspondence to:* Thomas C. van Leth (tommy.vanleth@wur.nl)

**Abstract.** Microwave links from cellular communication networks have been shown to be able to provide valuable information concerning the space-time variability of rainfall. In particular over urban areas, where network densities are generally high,

they have the potential to complement existing dedicated infrastructure to measure rainfall (gauges, radars). In addition, microwave links provide a great opportunity for ground-based rainfall measurement for those land surface areas of the world where gauges and radars are generally lacking. Such information is not only crucial for water management and agriculture, but also for instance for ground validation of space-borne rainfall estimates such as those provided by the GPM (Global Precipitation Measurement) mission.

The campaign described in this paper is dedicated to address several errors and uncertainties associated with such quantitative precipitation estimates in detail. The core of the experiment is provided by three co-located microwave links installed between two major buildings on the Wageningen University campus, approximately 2 km apart: a 38 GHz commercial microwave link, and 26 GHz and 38 GHz (dual-polarization) research microwave links. Transmitting and receiving antennas have been attached to masts installed on the roofs of the two buildings, about 30 m above the ground. This setup has been complemented with an

infrared large-aperture Scintillometer, installed over the same path, as well as 5 laser disdrometers and an automated rain gauge positioned at several locations along the path. Temporal sampling of the received signals was performed at a rate of 20 Hz. The setup is being monitored by time-lapse cameras to assess the state of the antennas as well as the atmosphere. The experiment has been active between August 2014 and December 2015.

Data from an existing automated weather station situated just outside Wageningen was further used to compare and to interpret

the findings. We find that a basic rainfall retrieval algorithm with no corrections already provides a reasonable correlation to rainfall as measured by the disdrometers. The microwave links do give a significant overestimation. We further investigate several events covering different attenuating phenomena: Rainfall, solid precipitation, temperature, dew, antenna wetting and clutter. We also briefly explore cases where several phenomena play a role. We conclude that the response of different makes of microwave antennas to many of these phenomena is significantly different even under the exact same operating conditions

and configuration.

## 1 Introduction

Accurate and real-time precipitation measurements are important for flood prediction, especially in urban areas. Traditional measurement techniques such as rain gauges have an insufficient temporal and spatial resolution to allow for accurate measurements in an urban setting (Berne, et al., 2004; Schilling, 1991). Furthermore, their spatial representativeness is limited

because of their small sampling areas, making them essentially zero-dimensional point measurements. Weather radars, in contrast, have a much larger sampling area making them more representative of the regional precipitation distribution, but their space-time resolution is often limited, in particular for urban applications. Furthermore, since they measure high up in the atmosphere (roughly 1000 meters), their measurements may not be representative of the situation near ground level. Finally, their high cost may be prohibitive for use by developing countries or local authorities.



Microwave link measurements may be a promising addition to the existing arsenal of rain measurement techniques. The use of such instruments for measuring precipitation was first suggested by Atlas and Ulbrich (1977). With respect to spatial representativeness and resolution, microwave links can fill some of the gaps between rain gauges and weather radar. The area sampled is along the path of the link: typically about a few kilometers long and a few meters to tens of meters wide at the

widest point. This makes the sampling footprint approximately one-dimensional. This is more spatially representative than a rain gauge, but less so than radar. However, microwave links have two major advantages over radar: they measure much closer to the ground than radar (typically a few tens of meters), and the relation between the measured variable (specific attenuation in the case of links and radar reflectivity in the case of radar) and rainfall intensity is much better defined and closer to linear for microwave links.

Despite these advantages, microwave links had not been deployed at a large scale since then, for the cost of setting up such a network would still have been quite severe. The real potential of microwave link measurements for precipitation measurement came with the realization that the microwave links used in cellular communications networks could be repurposed as precipitation measurement devices. This was demonstrated by Messer et al. (2006) and Leijnse et al. (2007a). This eliminates most of the cost of this technique as existing infrastructure can be used. This is especially valuable in developing countries,

which typically have few rain gauges let alone weather radar, yet do often have an extensive cellphone network (Doumounia, et al., 2014).

In the recent past there have been a number of studies towards the application of commercial microwave link networks for precipitation measurements. These studies have demonstrated the feasibility of this method in Southern Germany (Chwala, et al., 2012), the Netherlands (Overeem, et al., 2011; 2013), Israel (Zinevich, et al., 2008; 2009) and also in Burkina Faso

(Doumounia, et al., 2014) and Brazil (Rios Gaona, et al., 2017). Although the rainfall maps produced by this method show surprisingly good correspondence with the gauge-adjusted radar product (Overeem, et al., 2013), there are still inaccuracies remaining in the final products (Leijnse, et al., 2008; 2010; Zinevich, et al., 2010). Error sources can generally be divided into errors due to the mapping of the rainfall estimates from the microwave links, and errors in the individual measurements and the rainfall retrieval algorithm. It is in this last category where the largest remaining sources of error reside and not in the

mapping (Rios Gaona, et al., 2015). Therefore further research is needed into the microphysical aspects of the retrieval algorithm.

We present preliminary results from an experiment with three microwave links intended to investigate several possible sources of error in an urban environment in order to help fine-tune the existing retrieval scheme as laid forth in Overeem et al. (2011) and Overeem et al. (2016a) without resorting to simulated links from radar data. Several possible sources of error have been

identified previously: the wet antenna effect and related dew formation on antennas (Minda & Nakamura, 2005; Leijnse, et al., 2008), humidity (Holt, et al., 2003) and temperature (Minda & Nakamura, 2005), solid precipitation and spatial variability of precipitation (Berne & Uijlenhoet, 2007). Opportunities for simultaneous measurement of other environmental variables than rainfall have also been identified, such as evaporation (Leijnse, et al., 2007b), fog (Liebe, et al., 1989; David, et al., 2013), humidity (Chwala, et al., 2014) and hydrometeor type (Cherkassky, et al., 2014).

In this paper we describe a dedicated microwave link experiment that has been set up in the college town of Wageningen and analyze the results. The field experiment has been designed to provide validation data for microwave link rainfall retrieval at the scale of a single link, and to be able to compare different types of links simultaneously measuring along the same path. The goal of the analysis is to give a comprehensive overview of the phenomena encountered by a typical microwave link and to evaluate their relevance to a rainfall intensity retrieval. In order to do so we employ a relatively straightforward retrieval

algorithm with a minimum number of corrections and make use of a number of auxiliary measurement devices to gain insight into the retrieved signal. In section 2 a brief overview of the theoretical background pertaining to the operating principles of microwave link rainfall measurements is given. Section 3 covers a description of the experimental setup and the employed instruments. In section 4 the data processing methods applied in this experiment are detailed. In section 5 the obtained


experimental data is presented and an inventory of encountered phenomena is given. Finally, in section 6 conclusions are drawn.

## 2 Theoretical background

Both the attenuation of a microwave signal by rain drops during a rain event and the corresponding precipitation intensity can
be related to the rain drop size distribution. The precipitation intensity (in mm h$^{-1}$) can be calculated as follows, assuming the density of water to be constant:

$$R(t) = C_R \int_0^\infty V(D)v(D)N(D,t)\,dD \tag{1}$$

Where $V(D)$ is the volume of a raindrop in mm$^3$, $D$ is the raindrop diameter in mm, $v(D)$ is the fall velocity (in m s$^{-1}$) of a particle with diameter $D$ and $N(D,t)$ is the density of particles with diameter $D$ per m$^3$ or drop size distribution (DSD) as a
function of time $t$ and $R_C = 3.6 \cdot 10^{-3}$ is a unit conversion factor. When dividing the particle diameter into discrete classes (as is measured by a disdrometer) this can be approximated as follows:

$$R(t) = C_R \frac{1}{6}\pi \int_0^\infty D^3 v(D)N(D,t)\,dD \approx C_R \frac{1}{6}\pi \sum_{i=1}^n D_i^3 v(D_i)N(D_i,t)\Delta D_i \tag{2}$$

Here, $D_i$ is the mean diameter of the $i_{\text{th}}$ drop size class, $N(D_i,t)$ is the discrete drop size distribution. $\Delta D_i$ is the width of the $i_{\text{th}}$ diameter class, and $n$ is the number of drop size classes.

A similar function defines the specific (logarithmic) attenuation (in dB km$^{-1}$), where we assume that the particle density is low enough such that multiple scattering can be neglected:

$$k(t) = C_k \int_0^\infty \sigma_{ext}(D)N(D,t)\,dD \approx C_k \sum_{i=1}^n \sigma_{ext}(D_i)N(D_i,t)\Delta D_i \tag{3}$$

Here, $\sigma_{ext}(D)$ (in mm$^2$) is the extinction cross section of a hydrometeor with a diameter $D$ and $C_k = 100 \cdot \ln(10)^{-1}$ is a unit conversion factor. $\sigma_{ext}(D)$ is also dependent on the frequency and polarization of the incident radiation. It can be derived from
the forward scattering amplitude matrix $\mathbf{S}(D)$ which relates the incoming electromagnetic wave with the outgoing (forward scattered) wave:

$$\begin{pmatrix} E_h \\ E_v \end{pmatrix} = \mathbf{S}(D)\begin{pmatrix} E_{h0} \\ E_{v0} \end{pmatrix} = \begin{pmatrix} S_{hh}(D) & S_{hv}(D) \\ S_{vh}(D) & S_{vv}(D) \end{pmatrix}\begin{pmatrix} E_{h0} \\ E_{v0} \end{pmatrix} \tag{4a}$$

$$\begin{pmatrix} \sigma_{ext,h}(D) \\ \sigma_{ext,v}(D) \end{pmatrix} = \frac{\lambda^2}{\pi}\Im\left[\begin{pmatrix} S_{hh}(D) \\ S_{vv}(D) \end{pmatrix}\right] \tag{4b}$$

Where $\lambda$ is the wavelength of the radiation in mm, $S_{ij}$ is the element of the scattering amplitude matrix for the component of
radiation with incoming polarization $i$ and outgoing polarization $j$, where $v$ and $h$ represent the vertically polarized and horizontally polarized components, respectively (van der Hulst, 1957). The forward scattering amplitude matrix for spheres of arbitrary size and dielectric properties can be calculated with Mie scattering theory (Mie, 1908). In order to be able to calculate the scattering properties for non-spherical drop shapes, we make use of the T-matrix approach (Waterman, 1965; Mishchenko, et al., 1996).

The relations between the raindrop diameter on the one hand, and the raindrop fall speed, its extinction cross-section (and the differential phase) on the other closely resemble power laws (e.g. Atlas and Ulbrich, 1977). This means that both the specific attenuation and the precipitation intensity are approximately statistical moments of the DSD, which can themselves be empirically related by a power-law:

$$k = aR^b \tag{5}$$

Where $a$ and $b$ are fitted parameters (Atlas & Ulbrich, 1977) which are both dependent on the average DSD within the control volume.

The power law parameters derived for a small control volume are strictly speaking only valid for a larger volume when the rainfall intensity within that control volume is homogeneous. (Similar considerations apply for temporal aggregation.) When measuring path-integrated attenuation with path lengths typical for a cellular communication link, this is no longer the case.





Because the power-law relationships employed throughout the literature are all derived at the point scale (including this paper) the validity of those relationships at the path scale is therefore dependent on the near-linearity of Eq. (5) (that is: $b \approx 1$) at the relevant carrier frequencies, such that:

$$A = \int_0^L k(s)\, ds = \int_0^L aR(s)^b\, ds \approx aL\langle R \rangle^b \qquad (6)$$

## 3 Experimental setup

### 3.1 Global overview

The backbone of the experimental setup consists of three microwave links placed along the same path between two university buildings on opposite sides of the college town of Wageningen. As such, the majority of the 2.2 km long link path covers urban terrain (Fig. 1a). All transmitting antennas are placed on a two meter high mast at approximately 1.5 meter from the base of the mast (Fig. 1b). The mast is placed on top of a 7-story building. The building is situated atop a slightly elevated area on the south end of Wageningen (51.968657 °N, 5.68273 °E). The receiving antennas are placed on an identical mast on the roof of an 8-story building at the northern end of Wageningen (51.985230 °N, 5.664312 °E). The height above ground level is 27 meter on the transmitting end and 40 meter on the receiving end. The total height above sea level is 62 meter at the transmitting end and 51 meter at the receiving end. The terrain in between the endpoints of the path consists mostly of terraced housing, a sports field and other buildings of three stories or less. The maximum width of the first Fresnel zone (halfway along the path) at the featured frequencies is less than 5 meter thus considering the height of the antenna locations compared to the intermediate terrain, there are no permanent obstructions affecting the beam significantly.

The experiment has been operational from 22 August 2014 up to and including 8 January 2016. Not all instruments have been operational during this entire period though, as is indicated in Fig. 2. Also, from 7 August 2015 to 25 August 2015 all transmitters were nonoperational due to a local power outage.

### 3.2 Microwave and near-infrared links

Of the three links one is a Nokia Flexihopper, formerly part of a commercial cellphone network operated by T-Mobile. Such links are still used in cellphone networks around the world and this microwave link could therefore be regarded as representative of the link systems that would be used in an operational setting. The Nokia Flexihopper is set to transmit and receive at a frequency of 38.17625 GHz. The device transmits and receives only horizontally polarized radiation.

The other two links are custom-built by Rutherford Appleton Laboratories (UK) (RAL). The first operates at 26 GHz and transmits and receives only horizontally polarized radiation. It contains both a linear and a logarithmic detector. The second RAL link operates at 38 GHz. The receiver contains four detectors, two of which measure horizontally polarized radiation (linear and logarithmic) and the other measures vertically polarized radiation (idem). The phase difference between the horizontally and vertically polarized signals is measured by separate detectors as well. In this paper we will only deal with data from the logarithmic detectors. Note that the second RAL link measured at roughly the same frequency as the Nokia link. The frequencies are chosen to be far enough apart so as not to cause interference, but are close enough that the scattering characteristics of the radiation with respect to raindrops are almost identical.

A Scintec BLS900 near-infrared link is also placed together with the microwave links on the same path. It operates at a frequency of 340 THz (880 nm). This provides for comparison in the case of, for example, fog and other visibility-affecting phenomena. Similarly to the microwave links (despite operating in a different scattering regime), it could potentially also be used to measure rain intensity (Uijlenhoet, et al., 2011).

All link receivers are sampled with a Campbell Scientific CR1000 data logger and stored on a remote server on a daily basis. The sampling frequency is 20 Hz. Auxiliary data (e.g. operating temperature) is sampled at a frequency of 2 min$^{-1}$. The Nokia Flexihopper system consists of separate outdoor and indoor units, the latter containing the digital signal processing circuits





and power supply. Note that we have not actively used the indoor unit of the Nokia link system aside from the power supply; Instead, the analog detector signal normally used for automated gain control (AGC) is fed directly into the analog-digital converter (ADC) of the separate data logger. We do this to avoid the significant power quantization error (1 dB) that would be incurred using the link device's own AGC-ADC system. The analog signal was calibrated in an indoor environment using the

signal power indication of the indoor unit as a reference. The RAL links where recalibrated by Rutherford Appleton Laboratories shortly before the beginning of the experiment. The calibration curves are shown below in Fig. 3. The transmitted power for all devices was kept constant, but was not separately measured.

### 3.3 Additional instruments

To serve as a ground truth, we use OTT Parsivel laser disdrometers (Fig. 1c). These can measure not only precipitation intensity

but also the size and velocity distributions of passing precipitation particles over 30-s intervals. With this information they can provide an approximation of the type of precipitation that occurred. In this way it is possible to, for example, filter out solid precipitation from the microwave link data or select dry periods to determine the 'dry' baseline signal. Because of the small sampling footprint of these devices, they may not give a representative ground truth for the aggregated path measurements. Therefore five disdrometers are placed at four different locations, roughly evenly spread along the link path (Fig. 1a). At the

receiver end of the link path two disdrometers are placed next to each other in close proximity in order to test the accuracy of the disdrometers themselves. All disdrometers are placed on flat or gently sloping rooftops within Wageningen. The disdrometers all contain a built-in preprocessing unit which samples the raw laser amplitude signals, converts them to hydrometeor counts using an algorithm (undisclosed by OTT) based on the principle described in Löffler-Mang & Joss (2000) and aggregates the samples to 30-second intervals. One of the disdrometers at the receiver end has been operational since the

beginning of the experiment. It is connected to the same data logger as the link detectors. The other four disdrometers have been operational for a shorter timespan (see Fig. 2). They are each connected to a UMTS modem which relays the disdrometer data to a remote server in real time. See Jaffrain et al. (2011) for more details about these autonomous disdrometer stations.

At the receiver end of the link path an automated tipping bucket rain gauge is placed close to the two disdrometers (Fig. 1d), to provide an additional independent measurement. The gauge has a tipping volume of 0.1 mm. Two time-lapse cameras are

placed at each end of the link path. On each side one camera is pointed along the path and the other is pointed at the antennas themselves. These serve to allow visual inspection of the link path and the antennas, which can be useful for relating link behavior to physical events..

For the subsequent data processing we also make use of data from the nearby automated weather station "Veenkampen" situated roughly 2 km to the west of Wageningen (operated by the university's Meteorology and Air Quality group) for ambient

temperature and pressure measurements.

### 4 Data processing

### 4.1 Disdrometers

### 4.1.1 Preprocessing

The raindrop size and velocity distributions are corrected for known instrumental biases using the method of Raupach and

Berne (2015), which involves two steps. Step one is shifting the velocity distributions so that the average velocities per size class match the theoretical terminal velocities for raindrops of that size class. Step two is multiplying the number of detected particles per size class by a class- and rain intensity-dependent correction factor. These correction factors were obtained by Raupach et al. (2015) from concurrent measurements with a 2D video disdrometer (2DVD), assuming the 2DVD measurements to be unbiased. Using these corrected distributions we derive rain intensities and other bulk quantities.





Whereas Raupach et al. (2015) use the theoretical raindrop terminal velocity model of Beard (1977) to determine the bias in velocity distribution we use the model of Beard (1976). The former is a simplification and approximation of the latter, designed to reduce computational expense. However, we found that on a contemporary desktop computer the time needed to compute terminal velocities was negligible using either model. Both models need the ambient pressure and temperature to calculate the

raindrop terminal velocity. We used the temperature and pressure measured by the automated weather station "Veenkampen". Because this station is situated outside the built-up area of Wageningen, there might be a slight bias in temperature as compared to the urban areas that the disdrometers are situated in.

The model of Beard (1976) does not compute the terminal velocity directly from only the pressure and temperature but instead needs the density of the water drops and ambient air as well as the surface tension of the air-water interface as input. For the

density of water as a function of temperature we use the empirical formula of Kell (Battan, 1973). For the surface tension of the air-water interface we employ the empirical relation proposed by Vargaftik et al. (1983).

### 4.1.2 Derived data

In the subsequent analysis we compare the attenuation encountered by the microwave link signals with the precipitation along the link path. We also make use of a k-R relationship based on the actual precipitation along the path and the expected

attenuation due to this precipitation. In order to do so, we assume that the corrected drop size distributions obtained from the disdrometer stations represent the ground truth for that location. The specific attenuation is derived from the drop size distributions using Eq. (3) and Eq. (4). We derive values for a carrier frequency of both 38 GHz and 26 GHz and for both horizontally and vertically polarized radiation.

We calculate the scattering amplitude matrix for each diameter class using the T-matrix approach developed by Waterman

(1965). The computations are done using an algorithm adapted from FORTRAN code developed by Mishchenko et al. (1996; 1998; 2000) and reimplemented using the Python programming language. Since the laser disdrometer cannot provide information on the geometric shape or orientation of the particles, we make use of an orientation averaging scheme. For this purpose we have adapted the particle orientation averaging functions developed by Leinonen (2014) from their T-matrix package. The shape of the raindrops is approximated by an oblate spheroid, with axis ratio dependent on the volume-equivalent

diameter. We use the axis-ratios suggested by Thurai et al. (2007). The complex index of refraction is needed to calculate the T-matrix. For rain drops we assume the empirically determined formula for the temperature-dependent complex index of refraction for pure liquid water by Liebe et al., (1991) where we use a temperature of 15 °C. Precipitation intensity is calculated with Eq. (2), using the corrected drop size distributions.

All derived disdrometer quantities are then averaged over the link path using a weighted average over all five disdrometers.

For each point along the path, the value of the quantity is taken to be equal to the value derived at the nearest disdrometer. The mean over the path is therefore equal to the mean of the disdrometers weighted by the fraction of the path that is closest to that disdrometer. The precipitation type and presence as determined by the Parsivel algorithm are also used. In this case, the path-averaged type is assigned as 'mixed' whenever two or more Parsivels register different precipitation types. It is considered 'dry' only when all Parsivels agree that there is no precipitation. In all other cases when one or more Parsivels detect

precipitation, that precipitation type is assigned as the path-averaged value. We distinguish five broad categories of precipitation: liquid, snow, hail/ice pellets, graupel, and mixed/melting snow. In the subsequent analyses we will mostly be concerned with liquid precipitation, since the other types where rare during the observation period.

### 4.1.3 Attenuation-rainfall intensity relationship

The disdrometer-derived precipitation intensities and specific attenuations at the frequencies employed in the microwave links

are plotted with respect to each other in Fig. 4. Each dot represents a single 30-s disdrometer measurement (not path-averaged). Only data points that were characterized as liquid precipitation by the Parsivel algorithm and where precipitation intensity was





higher than 0.1 mm h$^{-1}$ where selected. These data were used to fit a power-law model using a non-linear least-squares algorithm. Goodness-of-fit for these relationships is very high: $R^2 = 0.956$ to $R^2 = 0.986$. Also note that the power-law exponents are all close to one, indicating that specific attenuation and rainfall intensity are nearly proportional to each other at the employed frequencies. These relationships can then be applied to the specific attenuations measured with the links.

### 4.2 Microwave links

The microwave link received signal levels were converted to rain intensities using a baseline algorithm closely following the algorithm used by Overeem (2011; 2013).

In order to calculate the rainfall intensities, the attenuation caused by precipitation and the attenuation caused by other atmospheric effects must be distinguished. Rahimi et al. (2003) proposed a two-step approach in order to do so.

The first step is to determine which of the sampled periods are dry. Overeem et al. (2013) used the assumption of spatial correlation of precipitation to determine 'wet' and 'dry' periods for microwave links in cellular communication networks. In short, a period is considered 'wet' if nearby links show a mutual decrease in received signal levels. As we are considering only a single path, such a method would not be relevant here. An alternative is to use the assumption of temporal correlation of precipitation. Schleiss et al. (2010) suggested using a moving window standard deviation threshold. Similarly, Chwala et al. (2012) used a Fourier-transform based method to distinguish between wet and dry spells. Other methods applicable to a single link path are e.g. a Markov switching algorithm (Wang, et al., 2012) and the use of dual-frequency links (Rahimi, et al., 2003). Here, the path-aggregated disdrometer data is used to determine dry periods independently of the microwave link data.

The second step in the algorithm is to determine a suitable baseline signal level using the selected dry periods. The implemented baseline algorithm uses a rolling median over all measurements classified as dry in the surrounding centered 24-hour period to determine the baseline signal for each time-step. The specific attenuation is then calculated as:

$$k = \max\left(\frac{Rx_{ref} - Rx}{L}, 0\right) \tag{7}$$

Where Rx is the received power and L is the path length. Precipitation intensity is derived from the corrected attenuation using the power-law relationship of Eq. (5). The parameters $a$ and $b$ in this equation are obtained from the disdrometer data as described in section 4.1.3.

The precipitation intensity is furthermore set to 0 when the disdrometer indicates dry weather. Note that we do not perform any a priori additional corrections on the microwave link precipitation estimate, such as correcting for wet antennas. The goal is, after all, to use this basic estimate to assess potential error inducing phenomena, not to evaluate a best effort estimation.

## 5 Results and discussion

### 5.1 Overview

In the following section, we use the rainfall intensity as measured by the Parsivel disdrometers as a reference to assess the link-derived precipitation. Unless stated otherwise, we use the corrected DSD-derived rainfall intensities, not the rain intensities that the internal Parsivel algorithm produces.

In order to better understand the different phenomena that contribute to the microwave link attenuation signal, we present a number of illustrative events from the dataset. We search for relatively unambiguous events in order to gain insight into the separate phenomena. We will first look at the performance of the simple algorithm for detecting liquid precipitation and take a quick look at solid precipitation. We will then show how temperature and dew formation at the antennas affect the signal. At the end, we will look at some currently unexplained phenomena and also give some examples where different phenomena occur simultaneously.





### 5.2 Rainfall events

We compare the link-derived rainfall rates using the simple algorithm (excluding any specific corrections) described in section 4.2 with the spatially averaged rainfall rates derived from the disdrometers using the corrected DSDs.

Figure 5 shows an example of a single short isolated rain event on 14 July 2015 as indicated by the disdrometers. We chose this example because there are no attenuation-inducing influences other than rain in this event. (Note that the received signal level of the Nokia Flexihopper is offset by 14 dB in order to fit into the plot. This is done consistently for all following figures) The event consists of two distinct small peaks. The first peak of the path-average rainfall intensity only reaches 0.7 mm h$^{-1}$, while the second peak reaches 8 mm h$^{-1}$. We see that the second peak causes a clear attenuation of the received signal level of all the links. The smaller peak in rain intensity causes only a small attenuation in the 38 GHz links, fully within the 95$^{th}$ and 5$^{th}$ percentile range of the signal level in the surrounding dry period, which we will regard as indicative of the background noise level. Although the presence of precipitation is detected unambiguously by all the instruments, the magnitude of the response differs between the links. Both the horizontally and vertically polarized detectors in the 38 GHz RAL link give very similar responses, which is expected as they receive different components of the same signal and also share a substantial part of their electric signal path, including the antenna itself. Most notable is the difference between the signal of the Nokia link and the RAL link operating at the same frequency and polarization. Although the magnitudes are similar, the Nokia link has far less variability of the baseline signal level than all other link instruments. This difference could be caused by the differences in internal electronics of the detector. Another point of interest is that using the median of all dry data points in the 24-hour period, our estimation of the baseline signal level of the 26 GHz RAL link and to a lesser extend the 38 GHz RAL link is too high, resulting in an additive overestimation of the rain intensity. To further illustrate the great uncertainty in determining the baseline signal level even in this relatively straightforward case, Fig. 5b also shows the rain intensity corresponding to a power level equal to the 5$^{th}$ percentile of the dry signal. The calculated apparent precipitation during the first peak is completely below this line, indicating that this can be regarded as noise. Regardless, the peak precipitation estimate from the Nokia link is very close to the disdrometer estimate.

We can also see that attenuation of the microwave link signal persists for several minutes after the end of the rainfall event (according to the disdrometers) and slowly decays during this time. This could be the consequence of the link antennas becoming wet due to the rain and subsequently drying up after the event (Minda & Nakamura, 2005; Leijnse, et al., 2008).

Also plotted in Fig. 5a is the received signal level of the near-infrared link. Attenuation of this signal is indicative of visibility. In this case the visibility loss seems almost entirely related to rain ($r = -0.86$).

We illustrate the response of the link signals to rain with two more example events of a longer duration. One low-intensity drizzle event and one higher-intensity convective rain event with some spatial heterogeneity. On both occasions we use only the times when the disdrometers indicate rain has occurred for further analyses.

The first event, on 24 November 2015, consists of a low-intensity drizzle period (intensities under 2 mm h$^{-1}$ for most of the event) that persists for around 12 hours. The course of the event is illustrated in Fig. 6. No other attenuating phenomena are found, although the RAL links show a gradual shift in the baseline power level over the course of the event. The Nokia link, on the other hand, stays remarkably stable during the entire event. Visual inspection suggests a reasonably close match in the patterns of precipitation. This is also reflected in Fig. 7a—d, which shows a reasonable correlation with disdrometer rain intensity for both the Nokia link and the 26-GHz RAL link (0.888 and 0.860 respectively) and less so in the 38-GHz RAL link (0.675 for vertical polarization and 0.437 for horizontal polarization). There is a strong overestimation of rain intensity from the links when compared to the spatial average of the disdrometers, both additive and multiplicative. Additive overestimation can be as high as 2.5 mm, which is more than the actual rainfall during most of the event. Additive overestimation is the least in the Nokia link-derived data, which seems to be in line with the very stable baseline. However, it is still 0.6 mm h$^{-1}$, which is a problem for accurately measuring accumulations from light rain events. We can also see that in this case visibility cannot be reliably used as a proxy for precipitation intensity as is most clearly seen after 13:00 hours.





The second event, on 4 November 2015, is a more spatially heterogeneous and probably convective rainfall event with higher rainfall intensities. The total event lasts for 8 hours (see Fig. 8). Peaks in spatially-averaged rainfall intensity during this event are on the order of 20 to 30 mm h$^{-1}$, and individual disdrometer measurements reach up to 55 mm h$^{-1}$. Once again the Nokia link is remarkably stable, while the 38-GHz RAL link has an uncertain baseline, especially in the early part of the event. During

most of this event visibility seems to be a reasonable proxy for the precipitation, but drops afterwards. Correlations of link-derived rainfall with disdrometer-derived rainfall are much higher overall (0.91 to 0.93) (Fig. 7e—h) than for the event on 24 November 2015. Additive bias is of the same order of magnitude as for the drizzle case, which means that the additive bias relative to the rainfall intensities is much less for this event than for the drizzle event. Multiplicative bias is roughly the same for all links (around a factor 1.3). The behaviour of the links between the two events is reasonably consistent.

We now compare rainfall intensities from links and from disdrometers for the entire measurement period; results are shown in Fig. 7i—l. Data points where the internal Parsivel algorithm indicated no precipitation or solid precipitation are excluded. We also exclude the period during which the link transmitters were not functioning. The Nokia link performs better than the RAL links in terms of correlations. In all cases the links significantly overestimate the rainfall intensity, both in an additive (regression intercept ranging from 0.6 to 2.0 mm h$^{-1}$) and a multiplicative (regression slope ranging from 1.3 to 1.7) sense.

There are also a number of outliers where the Parsivels register a very low amount of rain, while the links show a broad range of rainfall intensities. Note that this does not include cases where the rain as registered by the Parsivels is exactly 0 mm, because these data points are forced to zero in the link rainfall intensity retrieval algorithm. Both the general overestimation trend as well as the outliers could be attributed to attenuating phenomena other than rain being erroneously processed as rain in the basic algorithm. It is less straightforward to point out an a priori cause for the few cases where underestimation occurs,

but this could be due to uncertainties in the k-R relation that is used to retrieve rain. In typical operational settings larger temporal measurement intervals such as 15 minutes are common (e.g. Overeem, et al., 2016b). In order to illustrate the performance of a basic algorithm without any sort of correction at this resolution, Fig. 7m—p show the scatterplot and linear regression for the entire dataset, but down-sampled using a 15-minute mean. The correlation for the Nokia link is slightly lower with the 15-minute intervals than it is using 30-second intervals, yet the scatter around the regression line is also lower.

In the case of the RAL links, the performance is worse for the 15-minute accumulations than for the 30-s intervals.

## 5.3 Solid/mixed precipitation

The microwave link precipitation detection method is principally intended for liquid precipitation. Snow and hail have different electromagnetic characteristics (i.e. ice has a different refractive index than water, and the shapes of the particles are different). Therefore different attenuation-precipitation relations hold. Non-melting snow flakes cause very little attenuation in the

frequency range under study (e.g. Battan, 1973) and therefore we do not expect to be able to detect them. Wet snow hydrometeors, on the other hand, which consists of a mixture of solid and liquid water and air, generally cause more microwave attenuation than a raindrop containing the same amount of water. Since we are dealing with more complex shapes and multiple phases of water and air and therefore an inhomogeneous index of refraction, accurate estimates of wet snow attenuation and inversely, the estimation of snowfall magnitude through microwave attenuation, poses a real challenge (e.g. Paulson et al.,

2011). Nevertheless, we can still detect the presence of wet snow and melting ice pellets.

Very few solid or mixed precipitation events occurred during our measurement period. Figure 9 shows one of the few snowfall occurrences during the campaign, on 4 February 2015. At this point in the campaign only one disdrometer was yet placed and no rain gauge was available, which limits the potential for a quantitative comparison. Figure 9d shows time-lapse camera footage taken during different stages of this event. The background shades in Fig. 9b indicate the type of precipitation as

indicated by the Parsivel internal algorithm (blue is liquid precipitation, green is snow, red is mixed precipitation). The total event duration is about 40 minutes, yet the event is quite variable in time.





As indicated by the background colours and the camera footage, this short event starts out with a mixture of rain and ice pellets and then turns into snowfall. Along with the mixed precipitation the temperature drops from 4 °C to 1 °C. During the snowfall, the temperature drops further to 0 °C. The absolute values of the disdrometer-derived precipitation intensity should be taken with a grain of salt here, since our processing algorithm treats every particle as a raindrop. This results in far too high values

during snowfall, as we do not take into account the lower density of a typical snowflake. Because the disdrometer rainfall intensity shown here is that of only one disdrometer and since it was placed at one far end of the link path, we do not expect the small-scale variations to match exactly with those of the link attenuation.

Between 15:10 and 15:30, the links are attenuated with a magnitude that corresponds roughly with the precipitation intensity measured by the disdrometer, assuming that it is pure rain. Afterwards, when snow starts to fall between 15:30 and 15:55, the

precipitation intensity derived from the disdrometers becomes a factor 10 higher than the link-derived precipitation intensity, but this is likely to be due to the faulty disdrometer algorithm when applied in snow. While both the disdrometer and the camera footage seem to indicate that the precipitation stops after 15:55, the attenuation of the links persists until the signal level returns to its initial value between 16:00 and 16:15. At this point the temperature hovers at a few tenths of degrees above zero, and the camera footage indicates some residual snow is left on the antenna covers. The snow deposits are mostly on top

of the covers and is mostly still present by 16:17, when attenuation has decayed fully so snow deposits alone cannot explain the persistent attenuation.

Because there were few snowfall events during the entire campaign period and each of them was of short duration and mixed with other types of precipitation (similar to the event described in this section), no meaningful analyses could be done regarding the relationship between attenuation and snowfall intensity.

### 5.4 Temperature

Throughout the entire observation period a diurnal oscillation can be seen in the attenuation signal. This diurnal cycle is present in all signals, although the magnitude of the oscillation is in general significantly higher for the RAL links than it is for the Nokia link. The magnitude of the oscillation also varies throughout the observation period. This behaviour does not correspond

to any precipitation pattern but seems to follow the known diurnal variations in temperature. Although this pattern can be seen throughout the observational period, the correlation with temperature is not always clear, because the signal is generally much weaker when other attenuating phenomena are present.

We will therefore first focus on a relatively long dry period between the 14th and 24th of April 2015, as shown in Fig. 10. During this period the disdrometers picked up no precipitation; however, the received signal levels are not constant. Instead,

variations up to 1 dB are present. In Fig. 10b, the time series of ambient air temperature measured by the nearby weather station is plotted for the same period together with the visibility measured at that same station, while in Fig. 11, the power levels for this period are plotted against the temperature with a simple linear regression. We performed separate regressions for instances where humidity was above 90% and for instances where relative humidity was below 90%. Also indicated in Fig. 10a are the periods where relative humidity is above 90% (green shade) and the periods where the net radiation flux at the

surface is upwards (blue shade).

There is a very strong negative correlation (up to -0.92) between received power and temperature, especially when relative humidity is below 90%, and for the most part a linear regression makes for a good fit. This is true for all the link instruments, although the slope of the linear fit is much lower for the Nokia link (-0.024 dB K$^{-1}$) than for the others (between -0.1 and -0.2 dB K$^{-1}$). Even so, the residual square error of the Nokia link regression is also lower (0.082) than those of the RAL links (~0.3).

Apparently, the magnitude of the temperature dependence is very much specific to the make of the link, more so than to the frequency or polarization of operation. Similar temperature dependence of the signal was also reported by Leijnse et al. (2007) for a different microwave link instrument. The relationship between temperature and attenuation for the Nokia link breaks





down at high humidity. This phenomenon is probably related to dew formation at the antennas as is discussed in more detail in section 5.5.

In subsequent analyses we expand our investigation of a possible linear temperature dependency to the entire experimental period. However, we exclude two timeframes from this analysis. Firstly, the period between 6 August 2015 and 25 August

2015 when the transmitters where not functioning (but the receivers were). Secondly, we also exclude a period between 11 May 2015 and 19 May 2015 because a metal construction crane was positioned in the line of sight between the transmitter and receiver (see section 5.7) several times in this period.

We consider that dew-related wetting of antennas causes attenuation of the link signal, that this attenuation muddles the observed temperature dependency of the signal, and that this phenomenon seems only to occur when the nearby weather station

registers a relative humidity above 90%. Therefore, we must ignore the data points with a relative humidity above 90% in order to separate temperature effects from signal attenuation in the full time period. For the same reason, we also remove all data points where the Parsivel indicates any form of precipitation. We then find the correlations and slopes as indicated in Table 1. We see that the linear temperature dependence remains similar for the RAL links. However, the temperature dependence of the Nokia link is drowned out in the noise.

If we do not remove the data-points classified as rainy, the temperature dependence of the RAL research links is still roughly consistent. Even when we take only the data points classified as rainy, the results remain consistent.

## 5.5 Dew and fog

There is also another phenomenon apparent in Fig. 10, especially noticeable in the Nokia link: Some sharp drops in received power that evolve in the early morning and then quickly disappear again with peaks of 1 to 2 dB (see Fig. 10a). Comparing to

Fig. 10b it is clear that they do not coincide with any change in temperature. Instead, these peaks only appear in periods when the net radiation is negative and the relative humidity is above 90%. The power gradually returns again to the previous level when the net radiation becomes positive and the event is over as soon as the relative humidity drops below 90%. From Fig. 10 we can see that these instances (where humidity is above 90%) are not correlated to temperature. These characteristics indicate that dew formation on the antennas is a plausible explanation for this phenomenon. The hypothesis is as follows: relative

humidity in the air approaching 100% and a net loss of radiative energy at the surface are indicative of dew formation; water condenses on the antenna covers and builds up a thin layer of water which causes attenuation proportional to the thickness of the layer (see Leijnse et al., 2008); as the net radiative flux changes sign and the water layer dries up, the attenuation slowly returns to the baseline level.

Fog and dew often occur under the same conditions and it is therefore difficult to rule out fog as the principal cause, from

correlations alone. If we take the visibility as indicative of the amount of fog, then we can see that they indeed occur often (but not always) at the same time. However, the temporal patterns in the fog density within an event (Fig. 10b) are quite different than those of the detected attenuation (Fig. 10a). It is therefore most likely that this attenuation is caused by dew on the antennas.

In the case shown in Fig. 12 a different fog event is shown in more detail. In this case, time-lapse camera footage was available

for a significant portion of the event, and is shown in Fig. 12c. Here again, strong attenuation is experienced by all the links with a peak attenuation of 3 dB in the case of the Nokia link, yet none of the disdrometers detect precipitation. Therefore, it is likely that we are dealing with a different attenuating phenomenon than precipitation. Using the basic rainfall retrieval algorithm, this event would result in an accumulated rainfall depth of 26 mm. As the time series of attenuation is smoother than we would expect of precipitation, we are likely dealing with antenna wetting due to either dew formation, or the result of

fog. The effect of fog on microwave link attenuation has also been observed by e.g. Liebe et al. (1989) and David et al. (2013). However, it is debatable whether the underlying cause is the wetting of the antennas or the attenuation by the fog droplets themselves. Observation of the accompanying time-lapse camera footage (Fig. 12c) reveals a heavy fog during the early





morning which gradually clears up concurrently with the decrease in attenuation in the late morning. No camera footage was available at night during the increasing leg of the attenuation signal, because the cameras were set up to not record during low-light. However, comparison of the visibility data from the nearby weather station (Fig. 12b) with the pattern of attenuation seems to undermine a direct relationship with fog. Visibility measurements from the NIR link along the path itself is of limited

use in this case, since the fog is so heavy that the attenuation is 'saturated' for most of the duration. The striking correspondence of the sign switch of both net radiation and attenuation increase makes dew formation the most likely interpretation.

### 5.6 Wet antennas

Near the start of the measurement period a simple test case was performed to assess the effect of wet antennas on rainfall retrieval. During a dry sunny day (12 September 2014), while the ambient temperature was 21 °C, both the Nokia and the RAL

38-GHz link where artificially wetted in short bursts using a spray bottle. The antennas where wetted until saturation and then allowed to dry in the sun. In this way, the attenuating effect of wet antennas can be observed, decoupled from the attenuating effect of raindrops in air. The RAL 26-GHz link was not included in the test, as it was not yet installed at the time; however, the antenna cover design and material is identical to that of the RAL 38-GHz link (aside from its diameter) and therefore it is assumed that the effect is similar.

In Fig. 13a the resulting attenuation signal is shown. It is seen that wetting of one antenna of the Nokia link system can result in an extra attenuation of 3 to 5 dB, which is of the same order of magnitude as what is observed in dew and fog events (where presumably both antennas of a link are affected). This corresponds with a rain intensity of 15 to 22 mm h$^{-1}$ using the power law derived in section 4.1.3 (shown in Fig. 13b). The signal then follows an exponential decay pattern due to drying, with a decay time of 3 minutes. The RAL link response to wetting is completely different, which may be related to the way water

collects on the antenna cover surface. The extra attenuation due to wetting is only 1 to 3 dB and the decay has two distinct stages. The initial peaks drop in less than a second after the spray stops, with no discernible decay pattern. However, after the initial peak, the attenuation does not drop to the baseline level; It stays at relatively constant elevated level after the spray. After each new spray the level may or may not change; not necessarily to a higher level. Only 21 minutes after the last spray, which was administered shortly before 14:56, has the attenuation fully decayed to the dry level (the full length of the decay is

not shown on the graph). While the observations of the Nokia link conform to the theory of Minda and Nakamura (2005), which assumes a water layer of uniform thickness on the antenna, the observations of the RAL link do not. Figure 14 shows that, indeed, the assumption of a water layer of uniform thickness does not hold for the RAL link antenna cover. Instead of forming a smooth layer, the hydrophobic material of the antenna cover forces the water to either run off immediately or collect into a few large beads. The runoff leads to a reduced peak attenuation and an immediate drop afterwards, since the surface is

never fully covered in water. However, the bead formation leads to a long secondary decay time, since the reduced surface to volume ratio (as compared to a uniform layer) hampers evaporation. As recorded video footage shows, with each new squirt of water, some new beads form while some others grow and fall off. The number and sizes of the beads remaining afterwards is highly variable, which might give a tentative explanation as to why the secondary attenuation level changes after each burst (and can even become lower than the previous level).

### 5.7 Clutter

Figure 15 shows an example of a remarkable event that occurred several times in the observation period. Figure 12a displays the received power for the four microwave link signals in the period of 10 May 2015 to 12 May 2015. There is a sudden sharp signal decrease and 18 hours later a subsequent increase towards normal levels. The disdrometers do not indicate any significant rainfall event during this time. Inspection of the time-lapse camera footage (shown in Fig. 15b) indicates that a

large metal construction crane was positioned exactly in front of the link path during this time about 200 m from the receivers, while it is positioned differently and regularly moving outside this time period. A few hours later we see another momentary





drop in the received signal levels at which point the crane moves swiftly through the path. The Nokia link detects no signal loss during the long period, but it does on other similar occasions. Because the radius of the first Fresnel zone at this distance is only 1.3 m and the centres of the Nokia antennas are about 0.5 m higher than the centres of the RAL antennas, there is a distinct possibility that in some instances the obstacle was only blocking some of the links.

We see similar patterns on multiple occasions and each time a large metallic object was positioned in the path. On other occasions, for example, a window cleaners' metal gondola crane was the cause of the attenuation. These kinds of temporary obstructions of the link path cannot be ruled out in operational settings, and most of the time no continuous visual observations are available. Therefore, it would be advantageous to be able to recognise these signal patterns and remove them algorithmically.

**5.8 Compound phenomena**

There are several anomalies present in the dataset that cannot be easily explained by any other observed atmospheric phenomenon. It is important to take into account that there will always be unexplained anomalies.

Figure 16 shows another example of a rain event. It can be seen that the link attenuation signals mimic the temporal dynamics in the rainfall. It can also be seen that the RAL 38-GHz link shows continued attenuation after the first rain event has stopped.

One possible explanation could be because the antennas become wet themselves, which contributes extra to attenuation. However, the duration of the effect in this instance is almost 3 hours, which is somewhat inconsistent with the results from section 5.6 which suggests a duration in the order of 21 minutes. It is also inconsistent with other events during this experiment when only a short attenuation period was observed after a precipitation event. Here as well, after the second rain shower, no lingering attenuation is observed. The Nokia link shows no lingering attenuation, in both cases which is consistent with the

results from the wet antenna experiment. It is hard to specify why lingering attenuation effects occur after some rain events and not after others, but air humidity might play a role here. The relative humidity hovers around 90% after the first event, while it drops to 80% directly after the second event. It is even harder to specify why some antennas are much more affected by the phenomenon than others at a given time, however the beading effect of the wet antenna with hydrophobic antenna cover might be related to this.

The examples given in this paper up till now have been simple cases where rain and other attenuating phenomena occur in an isolated fashion. These cases are important to be able to investigate and explain these phenomena. However, many times throughout the investigated period multiple phenomena have occurred simultaneously, which is a complicating factor for retrieval algorithms. Figure 17 provides an example of a complex event occurring on 1 December 2015. In this case, there is a simultaneous light drizzle and fog. Figure 17c shows that the disdrometers register rain intensities of below 1 mm h$^{-1}$ over a

period of over 4 hours. Despite the low intensity, the drizzle does produce attenuation of the links between 10:00 and 13:00, as can be seen in Fig. 17a. Fog rolls in at around 13:00 as evidenced by the time-lapse footage (not shown here) and substantiated by the increasing relative humidity and decreasing visibility as seen in Fig. 17b. From 13:00 till roughly 14:30 fog and drizzle occur simultaneously and both contribute to the attenuation. The fog related attenuation is most likely the effect of additional wetting of the antennas. At 15:30 the fog has blown over or has dissipated. This is captured well by the Nokia

link attenuation signal. The RAL link signals remain attenuated until 20:00. This could be due to the antenna covers still being wet. As was pointed out in section 5.6, Due to bead formation, the hydrophobic antenna covers can stay wet much longer. Indeed, from 16:00 onwards, net radiation flux is away from the surface (indicated in the shading in Fig. 17b) and thus only wind drying can take place. The simultaneous occurrence of drizzle and fog could pose a problem for binary dew filtering algorithms such as the one proposed by Overeem et al. (2016b).



### 6 Conclusions

Microwave attenuation—rainfall intensity relationships were determined using 30-second integrated drop size distributions obtained from 9 months of data from five laser disdrometers (and another 9 months from one disdrometer). In Table 2, the determined parameters are compared with others found in the literature. The parameters found by Leijnse et al. (2010) were

based on drop size distributions collected in the Netherlands as well, but were collected using filter-paper in 1968 (Wessels, 1972). We also compare with the formal ITU (International Telecommunications Union) recommendation regarding the modelling of microwave attenuation due to rain (ITU-R Recommendation, 2005). We see that the exponents ($b$) are very similar for the relationships obtained in this work and those obtained by Leijnse et al. (2010) and the coefficients ($a$) found by Leijnse et al. (2010) are somewhat lower than those found here. We can also conclude that, the $a$ parameter is too low and the

$b$ parameter is too high in the ITU recommendation for the Dutch rainfall climatology. Considering the high quality of the fits and the large amount of data for a broad range of events used to produce these fits, we recommend to use these locally-derived power laws. We propose that analyses of disdrometer data from regions with different rainfall climatologies might be used to determine the universality of these parameter values. This pertains to the coefficient $a$ in particular, as the exponent $b$ will always be close to one for the employed frequencies (Table 2).

In this paper we have tested a straightforward rainfall retrieval algorithm applied to the microwave link measurements on the basis of the aforementioned power-law relationship and compared the results with five disdrometers positioned along the path. This allows us to assess what the quality of a retrieval would be without taking into account the effect of other sources of attenuation. It is seen that there is a strong overestimation of rainfall intensities by the microwave links when compared to the disdrometers, when no corrections for the phenomena discussed in this paper are applied. The response of the link signals to

liquid precipitation in terms of additive and multiplicative bias seems quite consistent over different types of rainfall. However, this means that drizzle is much harder to quantify than heavier rainfall events because there is an additive bias of roughly 0.6 mm h$^{-1}$ in the Nokia link and roughly 2 mm h$^{-1}$ in the RAL links, i.e., of the same order of magnitude as typical rainfall intensities in drizzle.

There are significant differences in the accuracy of the rainfall retrieval between the two different makes of microwave links

that we used, operating at the same frequency and polarization. In general, the commercial link has a less noisy and more unambiguously interpretable signal response than the dedicated research link. The latter overestimates the rainfall intensity more during pure rainfall events, and also exhibits a stronger unintended temperature response, leading to a less stable baseline attenuation in general. The commercial link does produce a stronger overestimation due to dew. The use of a hydrophobic antenna cover should in principle reduce overestimation due to wet antennas. However, in practice, it also leads to bead

formation which has adverse consequences. The beads take much longer to evaporate than a thin layer of water under similar circumstances, so after rainfall has stopped, or after dew conditions have subsided, the attenuation lingers much longer. More importantly, they make the magnitude of the attenuation during this drying-up period less predictable, because the configuration of the beads on the antennas is unpredictable. As such, we would tentatively recommend against the use of hydrophobic antenna covers for research links, although a more robust experiment might be needed to confirm this conclusion.

In general, the use of two different microwave links operating at the same frequency along the same path during the same time (which should theoretically produce the same results) resulted in two remarkably different signal responses to rainfall and other attenuating phenomena. Therefore, we recommend that, when making use of data from commercial networks, note should be taken of the specific manufacturers and models the network is comprised of and the retrieval algorithm should be optimized for those link devices. This is especially relevant when parts of the network are supplied by different manufacturers. The

remarkable stability of the Nokia link does, however, demonstrate the value of commercially available microwave links for precise rainfall measurements when sampled at high frequencies.

We have demonstrated the effect of several complicating phenomena in typical microwave attenuation data for rainfall retrieval. The collected data from this experiment could be used to assess the effect of different sampling strategies in



operational settings. The experimental data can also be used as a test dataset to improve existing operational algorithms and devise corrections for the plethora of attenuating phenomena described in this paper.

**Acknowledgements**

The Nokia Flexihopper link system was kindly provided by T-Mobile Netherlands. The OTT Parsivel disdrometers where provided by Alexis Berne and colleagues from the École Polytechique Fédérale de Lausanne (EPFL) in Switzerland. The funding for this research was provided by the former Netherlands Technology Foundation STW (project 11944). We want to thank Pieter Hazenberg for his critical contribution to installing the instruments and Henk Pietersen for his help in the preparation for this campaign.

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





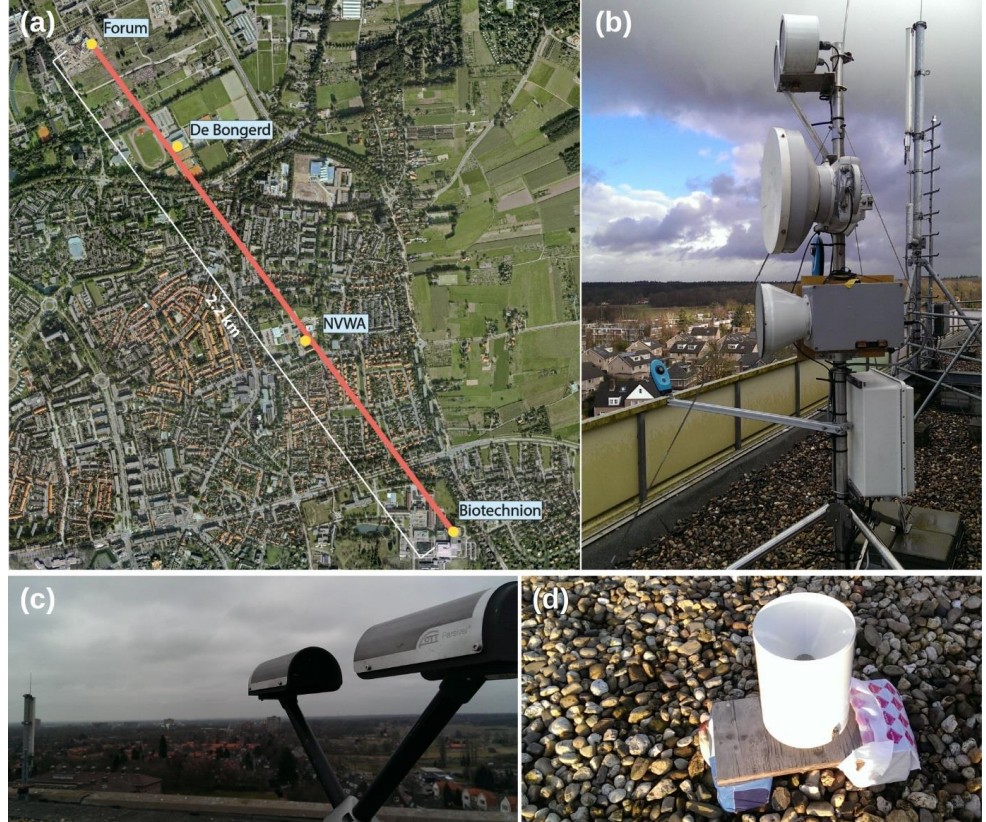

Figure 1: (a): A map of Wageningen showing the path of the links in red. The receiving antennas are at the end labelled "Forum"; the transmitting antennas are positioned at the end labelled "Biotechnion". The positions of the disdrometers are indicated with yellow dots. Each dotted position houses one disdrometer, except at the "Forum" position, where two disdrometers and an additional
5  tipping bucket rain gauge are placed. (b): The transmitting antenna mast placed on the roof of the "Biotechnion" building. From top to bottom: Scintec BLS900, Nokia Flexihopper, RAL 26 GHz (front), RAL 38 GHz (back). ((c): A Parsivel disdrometer (on the "Biotechnion" site). d) Précis Méchanique tipping bucket rain gauge at the "Forum" site.

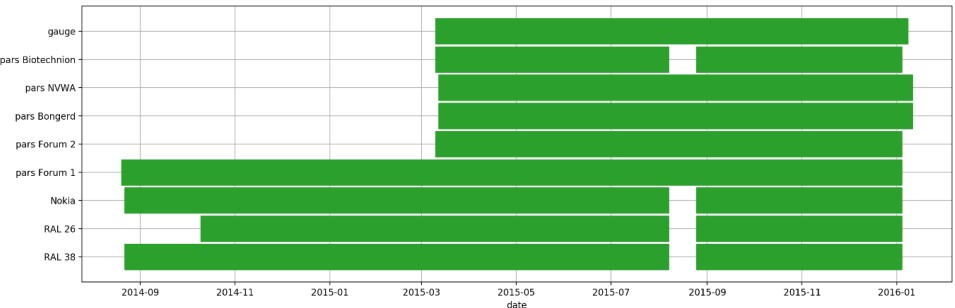

10  Figure 2: Operational period per instrument in the experimental setup.



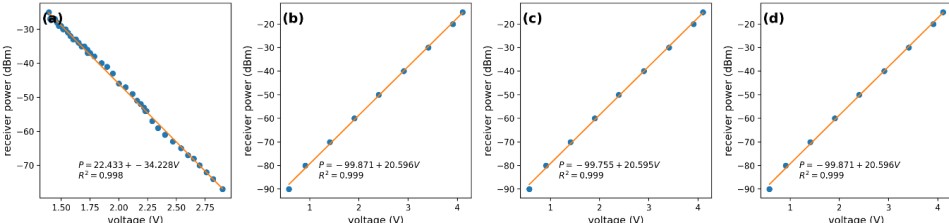

**Figure 3: Received signal power versus detector voltage read-out used for the calibration of the detectors. The orange line indicates the fitted calibration curve. (a) Nokia Flexihopper, (b) RAL 38GHz horizontal, (c) RAL 38 GHz vertical, (d) RAL 26 GHz.**

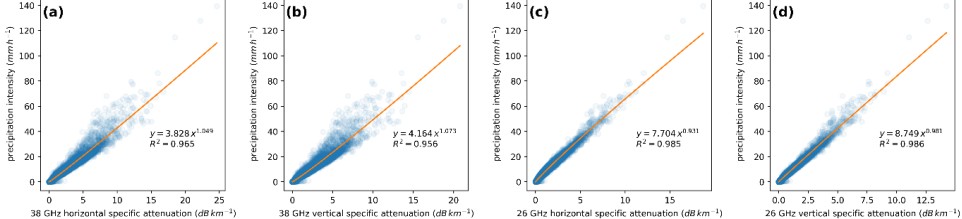

**Figure 4: Disdrometer-derived precipitation intensities plotted against disdrometer-derived specific attenuation at several frequencies and polarizations of the incident radiation. The Orange lines indicates the fitted curves. (a) 38GHz horizontal, (b) 38GHz vertical, (c) 26GHz horizontal, (d) 26GHz vertical.**

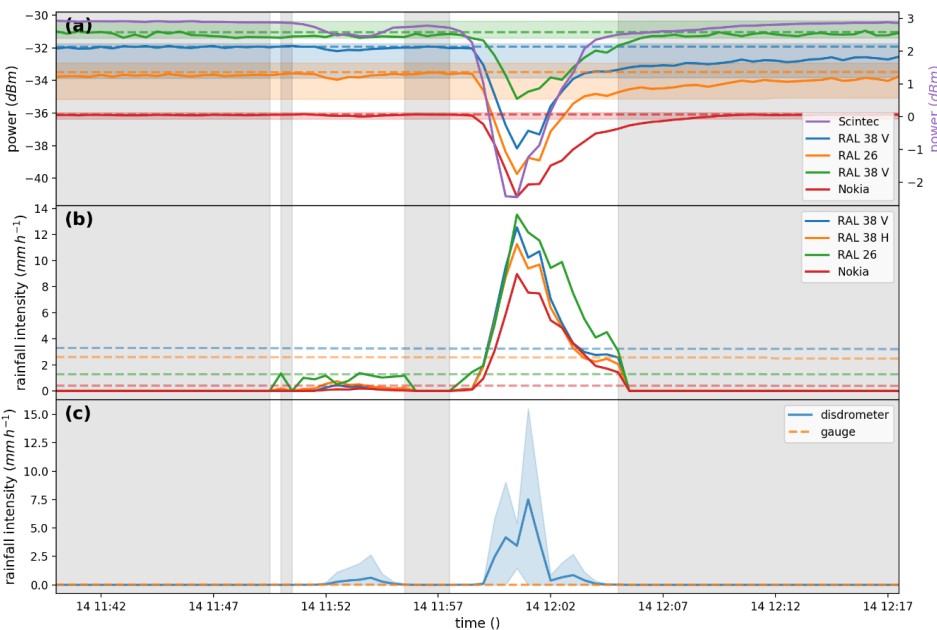

**Figure 5: Time series of an event on 14 July 2015. (a): Received power levels as solid lines, with the reference levels (median over dry periods in a 24-hour moving window) indicated in dashed lines. The 5$^{th}$ and 95$^{th}$ percentile power level over dry periods in a 24 hour moving window are indicated by the coloured shading. (b): Derived rainfall intensities using the basic algorithm in solid lines. dashed lines indicate the rainfall intensity resulting from applying the k-R relationship to the 5$^{th}$ percentile of the received power**

15 **levels in all dry periods in the 24-hour moving window. (c): The spatial weighted average Rainfall intensities derived from the disdrometers indicated by the blue line, with the weighted standard deviation among the disdrometers indicated with the light blue shaded area. The rainfall intensities derived from the tipping bucket gauge indicated with the dashed line. Dry periods, as indicated by the disdrometers, are indicated with grey shaded areas.**





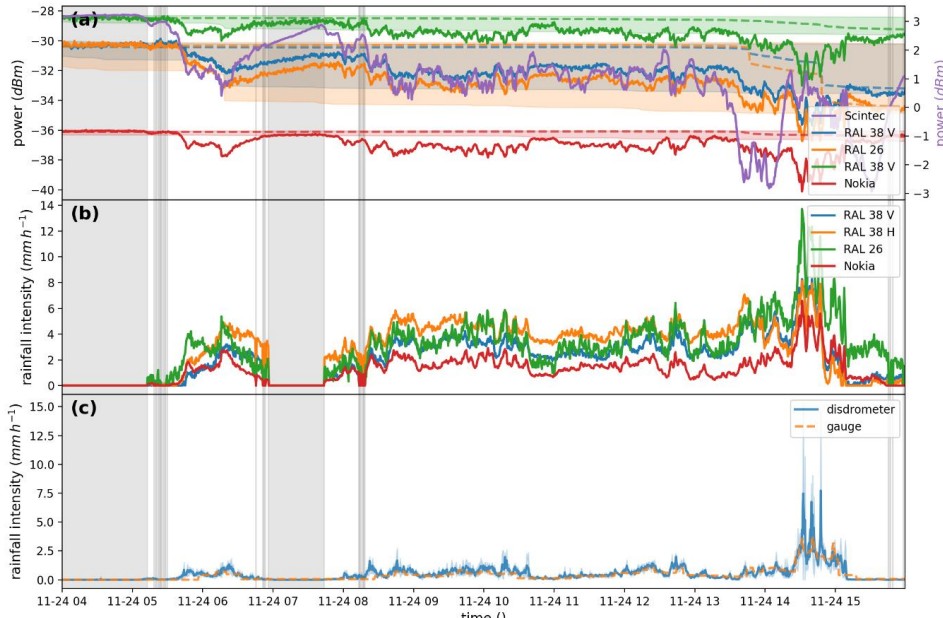

**Figure 6: Time series of an event on 24 November 2015. (a): Received power levels as solid lines, with the reference levels (median over dry periods in a 24 hour moving window) indicated with dashed lines. The 5th and 95th percentile power levels over dry periods in a 24 hour moving window are indicated by the coloured shading. (b): Derived rainfall intensities using the basic algorithm in solid lines. (c): The spatial weighted average rainfall intensities derived from the disdrometers indicated by the blue line, with the weighted standard deviation among the disdrometers indicated with the light blue shaded area. The rainfall intensities derived from the tipping bucket gauge indicated with the dashed line. Dry periods, as indicated by the disdrometers, are indicated with grey shaded areas.**





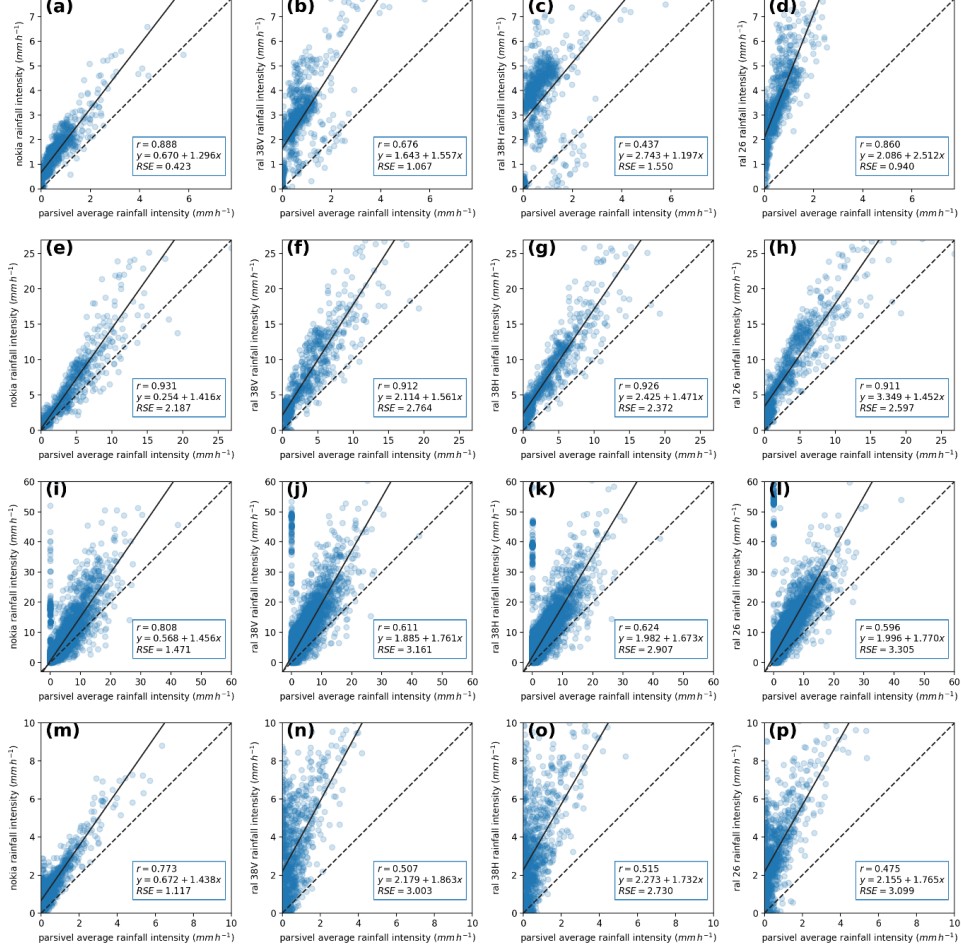

**Figure 7: Scatterplots of link-derived rainfall intensities versus disdrometer-derived rainfall intensities. Solid lines indicate a linear least-squares fit, dotted lines indicate the 1:1 line. Within each plot the correlation coefficient (r), the fitted line function and the residual standard error (RSE) are also shown. Links from left to right: Nokia, RAL 38GHz vertical, RAL 38GHz horizontal, RAL 26GHz. From top to bottom: 24 November 2015 (down-sampled to 30 s), 4 November 2015 (down-sampled to 30 s), whole dataset down-sampled to 30 s, whole dataset down-sampled to 15 min.**





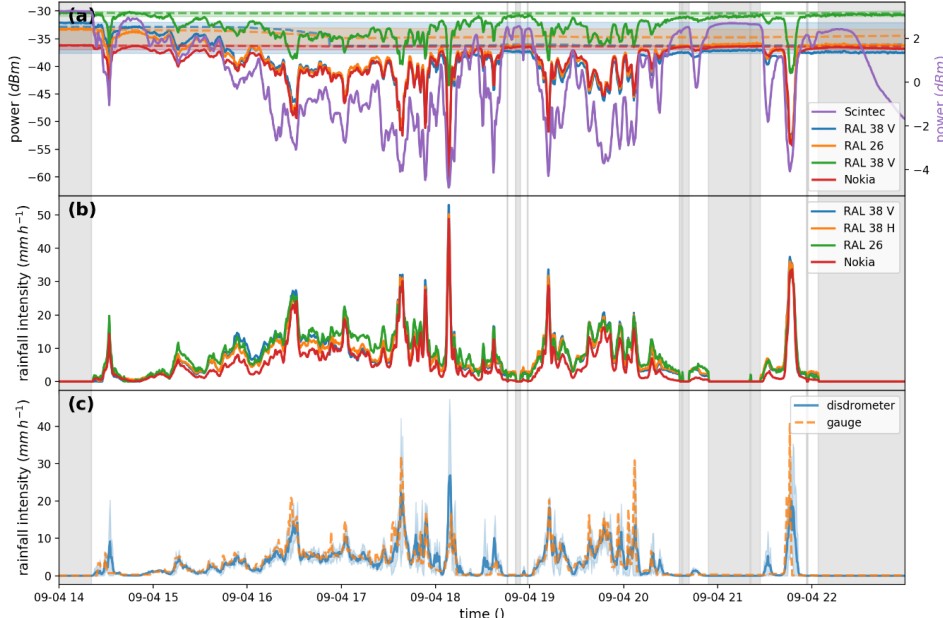

**Figure 8: Time series of an event on 4 November 2015. (a): Received power levels at the detectors as solid lines, with the reference levels (median over dry periods in a 24 hour moving window) indicated by dashed lines. The 5th and 95th percentile power levels over dry periods in a 24 hour moving window are indicated by the coloured shading. (b): Derived rainfall intensities using the basic algorithm in solid. (c): The spatial weighted average rainfall intensities derived from the disdrometers indicated by the blue line, with the weighted standard deviation among the disdrometers indicated with the light blue shaded area. The rainfall intensities derived from the tipping bucket gauge indicated with the dashed line. Dry periods, as indicated by the disdrometers, are indicated with grey shaded areas.**

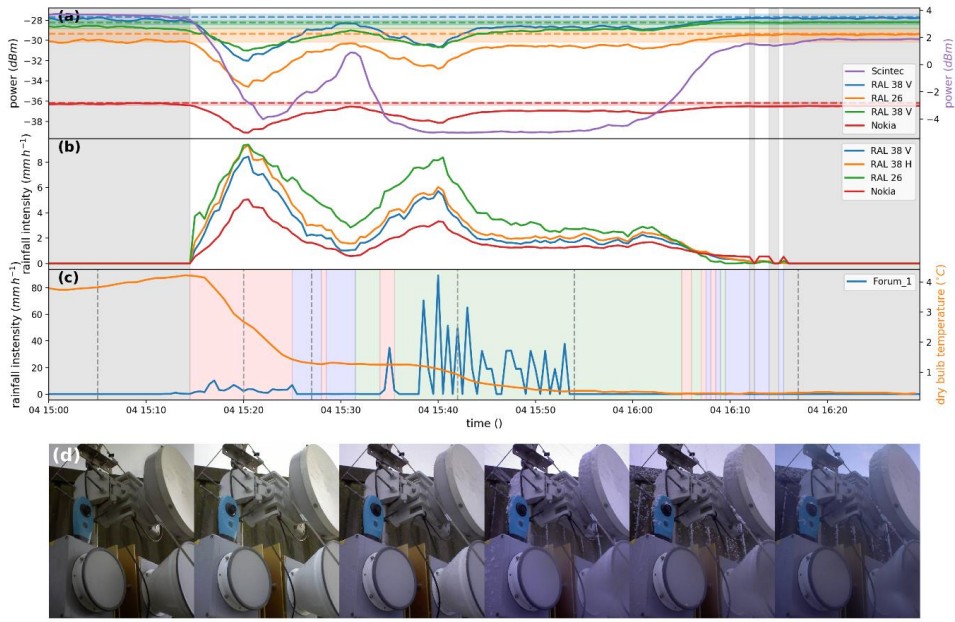

**Figure 9: Time series of an event on 4 February 2015. (a): Received power levels at the detectors as solid lines, with the reference levels (median over dry periods in a 24 hour moving window) indicated by dashed lines. The 5th and 95th percentile power levels over dry periods in a 24 hour moving window are indicated by the coloured shading. (b): Derived rainfall intensities using the basic**



algorithm in solid lines. (c): Rainfall intensities derived from the disdrometer positioned at "Forum" indicated by a blue line; ambient air temperature at 2 m at the "Veenkampen" meteorological station indicated by an orange line. Dry periods, as indicated by the disdrometers, are indicated with grey shaded areas. Periods with mixed precipitation are indicated with red shaded areas; periods where only liquid precipitation is detected are indicated in blue and periods with snow are indicated in green. (d): Images 5 from the time-lapse camera at the location of the transmitting antennas aimed at the antennas. The times at which these images where captured is indicated by the vertical dashed lines.

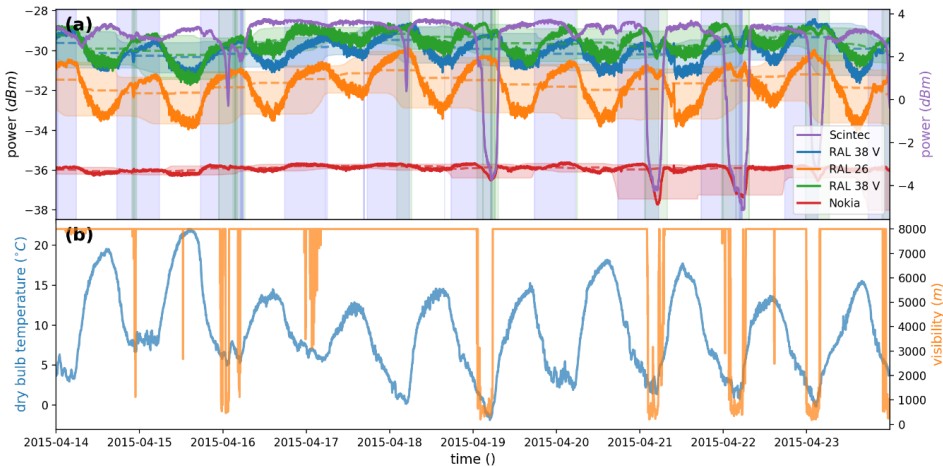

**Figure 10: Time series of the period between 14 April 2015 and 24 April 2015. (a): Received power levels as solid lines, with the** 10 **reference levels (median over dry periods in a 24-hour moving window) indicated by dashed lines. Periods with a negative net radiation flux at the surface are indicated with blue shading. Periods with a relative humidity >90% are indicated with green shading. (b): Several atmospheric variables measured at the "Veenkampen" meteorological station: visibility and ambient air temperature at 2 m indicated with orange and blue lines, respectively.**

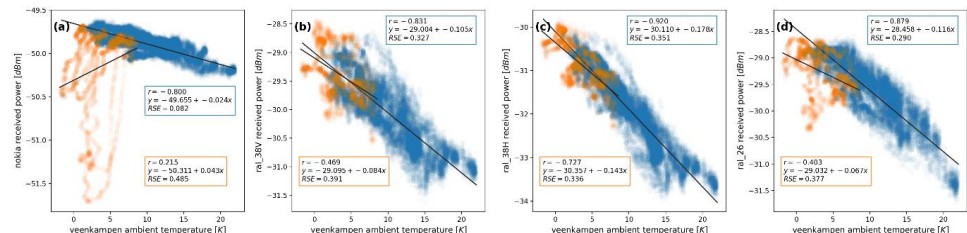

**Figure 11: Scatterplots of link received power versus ambient air temperature measured at the "Veenkampen" meteorological** 15 **station. Blue dots indicate times when relative humidity (as measured at "Veenkampen") is < 90%; orange dots indicate times when relative humidity > 90%. Solid lines indicate linear least-squares regression fit. Links: (a) Nokia Flexihopper, (b) RAL 38GHz vertical, (c) RAL 38GHz horizontal, (d) RAL 26GHz.**



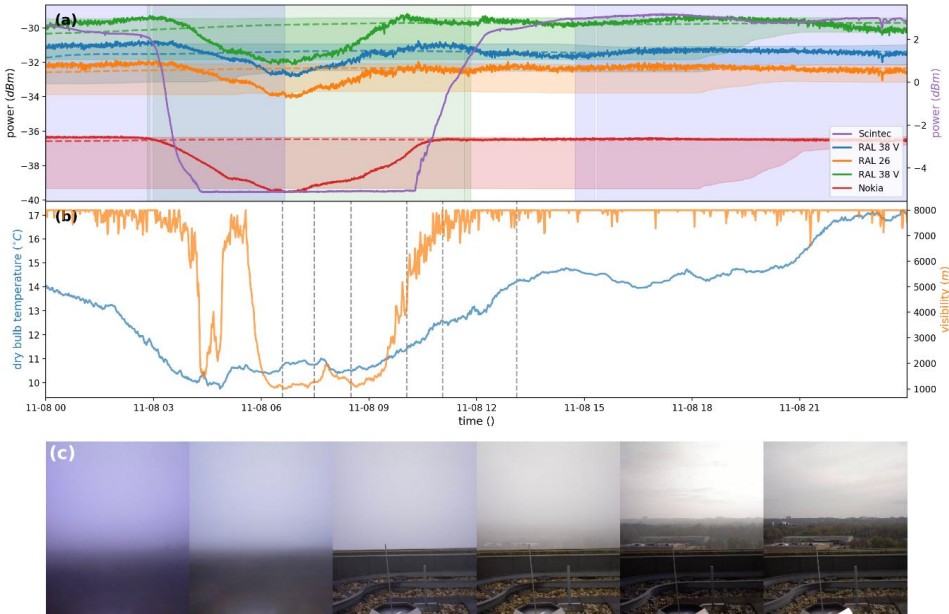

**Figure 12: Time series of an event on 8 November 2015. (a): Received power levels at the detectors as solid lines, with the reference levels (median over dry periods in a 24-hour moving window) indicated by dashed lines. Periods with a negative net radiation flux at the surface are indicated with blue shading. Periods with a relative humidity >90% are indicated with green shading. (b): Several atmospheric variables measured at the "Veenkampen" meteorological station: visibility and ambient air temperature at 2 m indicated with orange and blue lines, respectively. (c): Images from the time-lapse camera at the location of the receiving antennas aimed along the link path. The times at which these images where captured is indicated by the vertical dotted lines.**

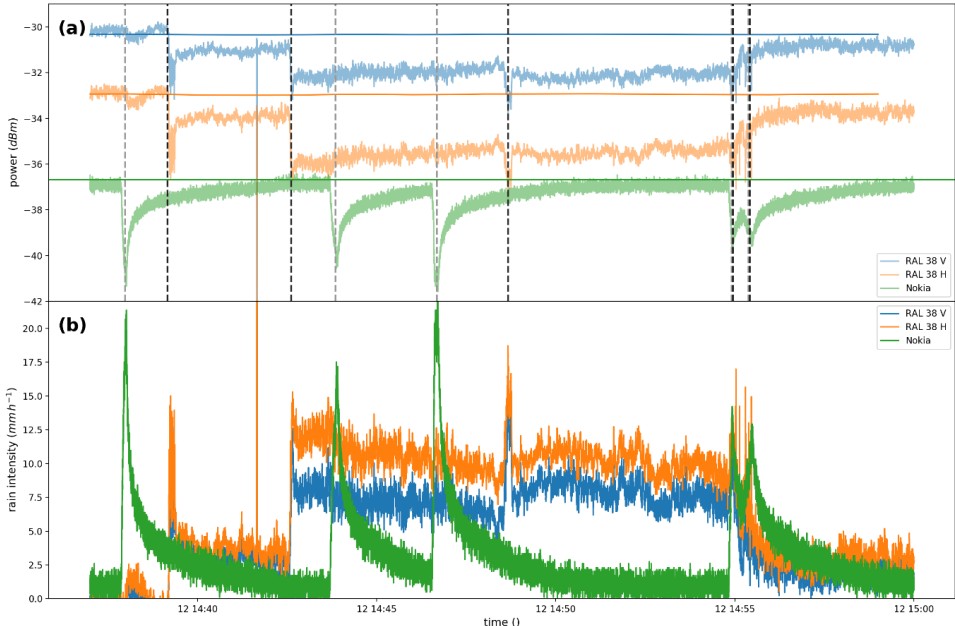

**Figure 13: Time series of the wet antenna experiment on 12 September 2014. (a): Received power levels at the detectors, with the reference levels indicated in darker hues. The reference levels are singular values manually fitted for this event. These are the raw 20-Hz sampled data, not the 30 second resampled data. The dotted vertical lines indicate the moments when a water spray was**





applied, with the dark grey lines indicating sprays on the RAL antenna and the light grey lines indicating sprays on the Nokia antenna. (b): Derived rainfall intensities using the basic algorithm.

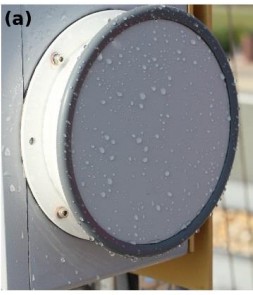
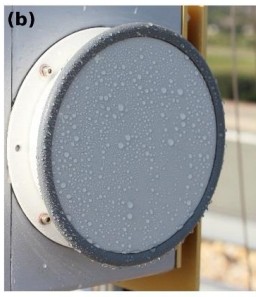
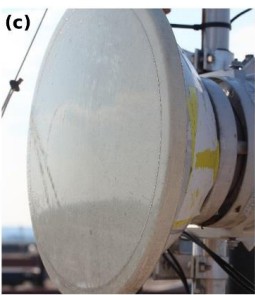

5    Figure 14: Photographs of the receiving antenna covers during the wet antenna experiment taken just after the antennas where sprayed. (a) and (b) are two instances of the RAL 38GHz cover. (c) is the Nokia Flexihopper cover.

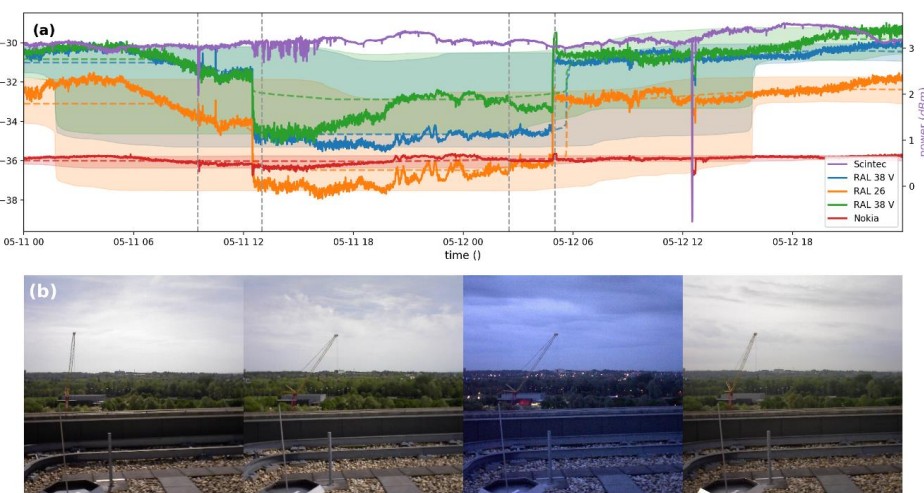

Figure 15: Time series of an event on 11 May 2015. (a): Received power levels at the detectors as solid lines, with the reference levels
10    (median over dry periods in a 24 hour moving window) indicated by dashed lines. The 5[th] and 95[th] percentile power levels over dry periods in a 24 hour moving window are indicated by the coloured shading. (b): Images from the time-lapse camera at the location of the receiving antennas aimed  along the link path. The times at which these images were captured is indicated by the vertical dotted lines.





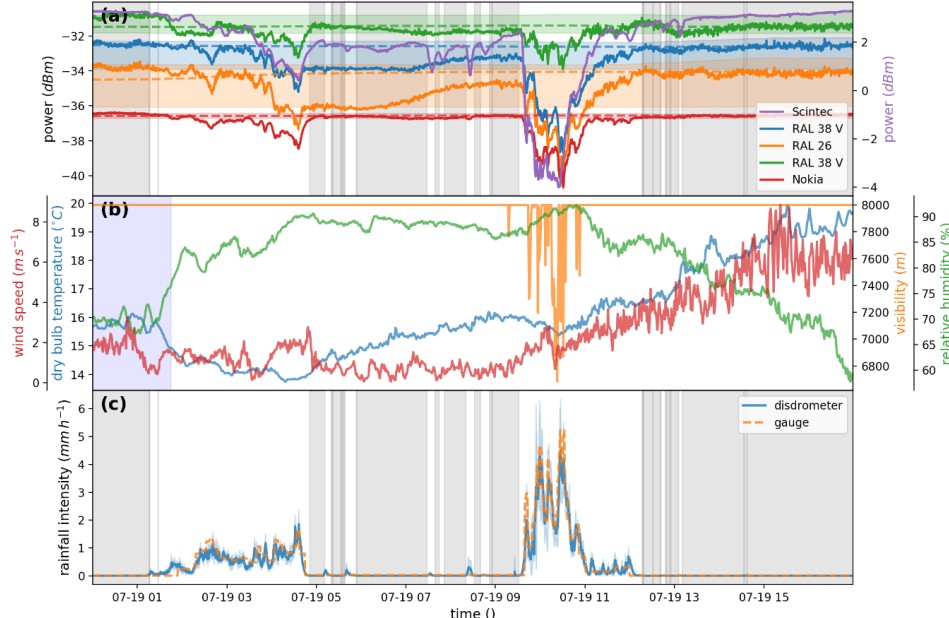

**Figure 16: Time series of an event on 19 July 2015. (a): Received power levels at the detectors as solid lines, with the reference levels (median over dry periods in a 24 hour moving window) indicated by dashed lines. The 5th and 95th percentile power levels over dry periods in a 24 hour moving window are indicated by the coloured shading. (b): Several atmospheric variables measured at the "Veenkampen" meteorological station: relative humidity, visibility and ambient air temperature at 2 m indicated with blue, orange and green lines, respectively. Periods with a negative net radiation flux at the surface are indicated with blue shading. (c): The spatial weighted average rainfall intensities derived from the disdrometers indicated by the blue line, with the weighted standard deviation among the disdrometers indicated with the light blue shaded area. The rainfall intensities derived from the tipping bucket gauge indicated with the dashed line. Dry periods, as indicated by the disdrometers, are indicated with grey shaded areas.**

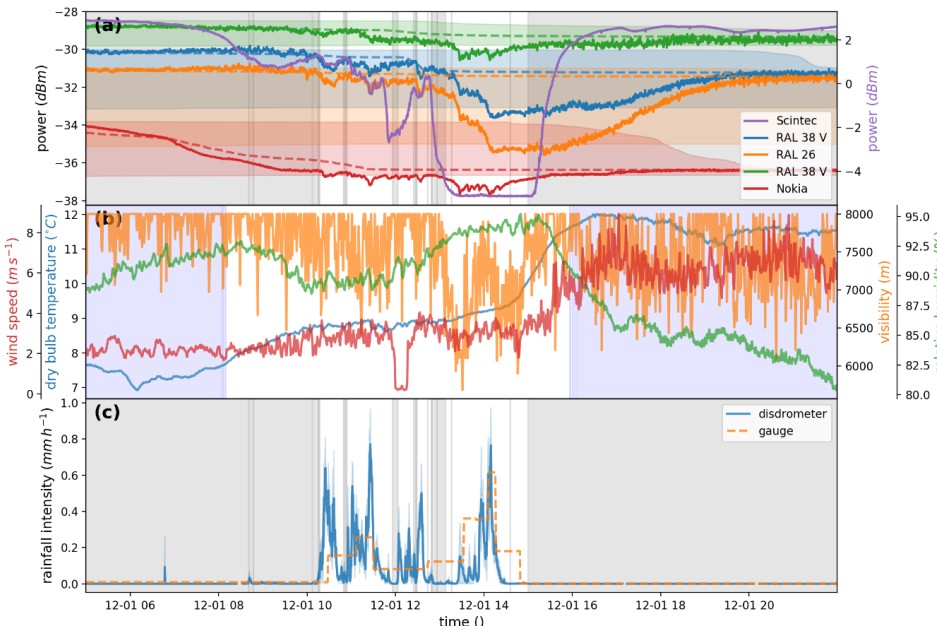

**Figure 17: Time series of an event on 1 December 2015. (a): Received power levels at the detectors, with the reference levels (median over dry periods in a 24 hour moving window) indicated in darker hues. The 5th and 95th percentile of power levels over dry periods**



in a 24 hour moving window are indicated by the coloured shading. (b): Several atmospheric variables measured at the "Veenkampen" meteorological station: relative humidity, visibility, ambient air temperature at 2 m and wind speed indicated with blue, orange, green and red  lines respectively. Periods with a negative net radiation flux at the surface are indicated with blue shading. (c The spatial weighted average Rainfall intensities derived from the disdrometers indicated by the blue line, with the weighted standard deviation among the disdrometers indicated with the light blue shaded area. The rainfall intensities derived from the tipping bucket gauge indicated with the dashed line. Dry periods, as indicated by the disdrometers, are indicated with grey shaded areas.

Table 1: results of the regression of Fig. 11 applied to different subsets of the data.

|  | Corr. Nokia | Slope Nokia | Corr. RAL 38V | Slope RAL 38V | Corr. RAL 38H | Slope RAL 38H | Corr. RAL 26 | Slope RAL 26 |
|---|---|---|---|---|---|---|---|---|
| **14—24 April** | -0.800 | -0.024 | -0.831 | -0.105 | -0.920 | -0.178 | -0.879 | -0.116 |
| **Whole set** | 0.019 | -0.003 | -0.461 | -0.153 | -0.565 | -0.179 | -0.546 | -0.113 |
| **Rain only** | 0.011 | 0.004 | -0.332 | -0.170 | -0.342 | -0.170 | -0.408 | -0.123 |
| **Dry only** | -0.072 | -0.001 | -0.573 | -0.168 | -0.716 | -0.197 | -0.719 | -0.134 |

Table 2: Coefficients and exponents ($a$ and $b$ parameters) of the R–k relationship derived from different sources for frequencies of 38GHz and 26GHz for both horizontally and vertically polarized radiation.

|  | $a_{38H}$ | $b_{38H}$ | $a_{38V}$ | $b_{38V}$ | $a_{26H}$ | $b_{26H}$ | $a_{26V}$ | $b_{26V}$ |
|---|---|---|---|---|---|---|---|---|
| **This paper** | 3.83 | 1.05 | 4.16 | 1.07 | 7.70 | 0.93 | 8.75 | 0.98 |
| **Leijnse, 2010** | 3.35 | 1.02 | 3.70 | 1.05 | 6.72 | 0.91 | 7.79 | 0.95 |
| **ITU-R** | 2.82 | 1.13 | 3.06 | 1.17 | 5.92 | 1.01 | 6.69 | 1.06 |