# Peer review of "A measurement campaign to assess sources of error in microwave link rainfall estimation"

_Atmospheric Measurement Techniques, 2017_

## Referee Comment (RC1) · Anonymous Referee #1 · 1 Jan 2018

General comment:

The manuscript describes dedicated experiment designed to investigate different phenomena influencing rainfall retrieval from microwave links. Several microwave links were installed over same path and equipped with time lapsed cameras shooting antenna surfaces and the link path. In addition, array of disdrometers completed with rain gauges were placed along the link path. Finally, additional observations form nearby weather station such as temperature, humidity or wind speed were used to interpret phenomena occurring during the measurement campaign. The manuscript goal is to provide comprehensive overview of different phenomena causing attenuation of microwave links and evaluate their relevance for rainfall intensity retrieval, specifically to the rainfall retrieval algorithm as suggested by Overeem et al. (2011 and 2016). The

first goal is scientifically relevant as i) it might improve understanding of uncertainties affecting microwave link rainfall retrieval and ii) description of attenuation patterns from other phenomena than rainfall is crucial for improving baseline separation algorithms. The presented experimental setup is very well suited to provide reliable dataset to reach this goal. The second goal is bit too specific to the selected processing algorithms (Overeem et al. 2011 and 2016).

The manuscript focuses on describing different phenomena causing link attenuation on several selected events. Overall statistical evaluation is mostly not provided which hinders quantitative assessment of the influence of these phenomena on microwave link rainfall retrieval. Results are often presented qualitatively in subjective manner (e.g. 'link is remarkably stable') even in cases where it could be easily described quantitatively, for more details see specific comments. Authors should distinguish in the whole result section more properly if the attenuation occurs along the link path or if it is rather related to hardware of microwave link radio units/antennas. The ambiguous cases should be then properly discussed and possibly confronted with radio wave propagation theory or results of other studies.

The manuscript is well structured, however, stylistics might be still improved, e. g. paragraphs in the result section could be more concise and fluent.

Specific comments:

P7L28: Results and discussion section: The results of microwave links are in the text mostly presented in mm/h although figures show also dBs. I strongly recommend to present the results also in dBs and compare them with theoretical rain induced attenuation from disdrometer data (eq. 3). The main reasons are these i) the uncertainties arising from imperfect separation of rain-induced attenuation are mixed with uncertainties arising from rainfall-attenuation power-law model, i.e. variability of $\alpha$ and $\beta$ parameters (Tab. 2) during different rainfall events and uncertainties due to path-integration of attenuation and nonlinearity of power-law model. This hinders interpretation of results.

ii) Substantial part of link attenuation unexplained by raindrops are hardware related errors (e.g. due to wet antenna or quantization noise). Such uncertainties expressed in mm/h apply only to links of the same lengths as in the experiment. iii) Most of the literature concerning microwave link propagation and different phenomena influencing radio wave attenuation (including wet antenna attenuation) express results in dBs.

P8L5: It is stated here that in the presented event there are 'no attenuation-inducing influences other than rain', however, this is inexact as the radio waves are during this event for sure attenuated e.g. by atmospheric gases, there is a free space loss, etc.

P8L33: There are certainly various attenuating phenomena (see comment P8L5) influencing link attenuation, and drop down in the RAL link signal level has probably some (uknown?) reason.

P9L4: 'remarkably stable' or 'uncertain baseline' is very subjective description. Please quantify.

P9L17-20: The causes of outliers and overestimation discussed in these lines are speculative. The experimental design should enable investigate unexpected behavior of links much more specifically thanks to reliable ground truth, cameras, etc. For example, it is stated here that 'overestimation and outliers could be attributed to attenuating phenomena . . . erroneously processed as rain in the basic algorithm'. It should be, however, possible to check against disdrometer data if the errors are due to the processing algorithm. Similarly, errors introduced by k-R model can be estimated and it should be verified if they can explain underestimation.

P10L22-24: Please quantify the magnitude of oscillations.

P10L33: Is the 90 % humidity threshold selected arbitrary, based on radiowave propagation theories, or estimated by regression itself? Please indicate.

P11L13-15: The statement that 'the temperature dependence of the Nokia link is drowned out in the noise' is speculative as you cannot prove there is a temperature

dependency if it is 'drowned out in the noise'. If you can prove it (at reasonable confidence level) it is then not 'drowned out in the noise'.

P11L19: Please indicate in the text the duration of antenna wetting and drying quantitatively. The figures depict too long period to distinguish if the processes take place only few minutes, tens of minutes or few hours.

P11L31-32: Please describe more precisely what is meant with 'quite different pattern'. Different range, variability, autocorrelation structure, . . .?

P13L11-12: What is meant with 'any other atmospheric phenomena'? Furthermore, the following text relates the attenuation to the humidity which is an atmospheric phenomena. The whole meaning of this sentence is, therefore, unclear.

P13L15-22: The antenna drying times might be very much influenced also by other environmental variables such as wind or sun radiation. Could e.g. wind which is also displayed in the figs 16 and 17 explain part of the uncertainty in drying duration? Is there any reason why humidity is included in the quantitative analyses and not the wind?

P14L20: A robust evidence that the link response to the additive and multiplicative bias is consistent over different events has not been provided in the previous text. Why don't you e.g. quantify both additive and multiplicative bias for each event and link and provide information about range and variability of both types of biases?

Figures: There is a wrong legend in the panel (a) of the figures 5, 6, 8, 9, 10, 12, 15, 16 and 17 as RAL 38 V is assigned to both blue and green lines. It seems to be that green line belongs to the RAL 38 H and the orange one to the RAL 26 V, i.e. same coding as in the panel (b).

References:

Overeem, A., Leijnse, H. and Uijlenhoet, R.: Retrieval algorithm for rainfall mapping from microwave link in a cellular communication network, Atmos. Meas. Tech., 9,

2425-2444, doi: 10.5194/amt-9-2425-2016, 2016.

Overeem, A., Leijnse, H. and Uijlenhoet, R.: Measuring urban rainfall using microwave links from commercial cellular communication networks, Water Resour. Res., 47, W12505, doi: 10.1029/2010WR010350, 2011.

---

## Referee Comment (RC2) · Anonymous Referee #2 · 10 Jan 2018

The authors have put together an experimental campaign in order to explore in depth some aspects of rainfall measurement from microwave links attenuation and understand better the uncertainties, focusing an a relatively short links (2 km) and medium high frequencies (26/38 GHz). The experimental set up is impressive and thorough, with 2 microwaves links –one operating at 38 Ghz and the other one dual frequency 38/26 GHz and dual polarization- sharing the same (2 km long) path ; 5 disdrometers in order to analyze rainfall intensity and microphysical variability along the path and at both ends, and an additional rain gauge ; a near IR scintillometer, cameras and a met station complement the set up and provide additional information on visibility and other atmospheric variables that might influence or help understanding the MWlinks signal fluctuations.

This is an ideal setting to analyze and quantify – at least for the 2 frequencies and the path length that are available here – some of the uncertainties in MWlinks based Quantitative Precipitation Estimation : -variability and time/space scale issues in the k-R relationship -wet antenna attenuation -baseline derivation uncertainty and non-rain induced fluctuations of the signal

-Additionally the MWlink power level is sampled at 20 Hz and with a small quantization error - which could be used to investigate errors due to coarse quantization and to the subsampling of the signal typically provided by Commercial MWLinks network monitoring systems (only providing max and min power every 15 minutes is common).

-Also the dual-frequency and dual-pol capability, together with the 5 disdrometers will allow to go further than the simple k-R based retrieval.

First of all I would like to congratulate the authors for the experiment they have put together and acknowledge the amount of work and time which will be necessary to fully exploit such a data set !

The main objective of the present paper is to present the experimental setting itself and some preliminary results which illustrate -in a rather qualitative manner- some of the issues that could be further explored with the data set : the discrepancy between the rainfall retrieved by the links and the path average rainfall retrieved by the 5 disdrometers, illustrated with 3 rainfall events ; some illustration of measurement during mixed precipitation ; effects of temperature on the signal fluctuations; wet antenna attenuation and its sensitivity to the type of radom material ; effect of dew and fog ; effect of clutter. Altogether an interesting catalog that illustrates the complexity of mwlink based retrieval of precipitation is provided in section 5. However, the reader stands a bit frustrated by a somehow QUALITATIVE OVERVIEW OF VARIOUS CAUSES OF MW SIGNAL FLUCTUATIONS, WITH A LACK OF STATISTICAL AND QUANTITATIVE ASSESSMENT OF THEIR IMPACT ON I) DETECTING/QUANTIFYING ATTENUATION DUE TO RAIN AND II) RETRIEVING RAIN RATE.

I would suggest that Sub-section 5.2, which has the most quantitative results and focus on the main objective of the MWlinks exploitation, i.e. rain retrieval, become a full section and be improved with some more quantitative results.

The other sub-sections in 5 should be lightened (5.3 and Fig9 which is essentially qualitative can be suppressed) and more focused on explaining some of the discrepancies observed in 5.2.

The text itself needs revising ; some expressions or comments are more subjective than scientific – and the authors sometimes overgeneralize their statements e.g. P4 L1 'the power law in the literature are ALL derived at point scale ' P 13 L12 'it is important to take into account that there will always be unexplained anomalies' P2 l 39 :' a relatively straightforward algorithm' P3 L31 'the relations . . ... closely resemble power law' etc. . . see more below.

DETAILed SUGGESTIONS : Title/Introduction :

- The stress on urban in the title is misleading – the paper does not focus on an urban problem or urban hydromet scales specially. A tile like 'A multi-instrument microwave link measurement campaign' would be better. - The introduction also stresses a lot on urban scales which is not that relevant since a single link and not a dense network are studied here. - P1 L33-L35 confusion between the space/time resolution of a single gauge and the problem of gauge network density versus scale of phenomenon. - P2 l8 : modern radar also used propagation variables such as Kdp and not just Z. . .. - P2 L25 : 'therefore further research . . ... microphysical aspects' – Sentence not clear + microphysics is not really the focus of this paper. . ... - P2 L 27 : relevance of urban ? ' in order to help fine tune the existing retrieval algorithm' - The sentence is clumsy and there is nothing about tuning the algorithm in the presented work anyway. . ... L 29 : simulating links from radar data is not at all relevant to what is proposed here and to the local scale ( one single 2 km path) studied.

Section 2 – P3 L 29 ' the relations . . ... Resemble power laws' – Paragraph to be revised

– false or approximative statements.

k-R relationship discussion : the discussion on k-R is spread in different parts of the manuscript with no consistency . The paragraph starting on P3 L35 is rather confused. There seem to be a confusion between DSD/rain type variability between rainfall events and k-R variability as a function of the scale considered (point versus path. . .). The concept of 'control volum' is unexplained and unclear. P3 Eq(6) insists on the problem of linearity of the relationship and the problem of point versus path average k-R relationship - but the fact that in this work the path average k-R relationship is effectively derived thanks to 5 disdrometers in not mentioned. . . . Subsection 4.1.2-4.1.3 should be merged and with a more explicit title like 'deriving path averaged k-R relationship'. Also note that k=aRˆb is used in (5 and 6) while a and b in Table 2 are for R=a kˆb most confusing

Section 3 : P4 L25 – please give the same precision for the frequency of the Nokia and RAL links. L24 : 'representative of THE link systems that would be used . . . . ' to be rephrased carefully – not all CMLs are Nokia and you dont use the NOKIA sampling/digitalization. . . . P4 L 31 : 'roughly' – unprecise L32-33 give the exact frequencies.

Section 4 4.1.2/4.1.3 – merge and improve. It seems that you are deriving a path-averaged k-R relationship based on weight values of both k and R derived from the 5 disdrometers with 30s long DSD spectra. Is this the case ? not very clear from the text. This very important point should stressed : most studies do not have access to the path average k-R and have to infer it from ponctual k-R and assumptions on rainfall variability . The differences between the single disdrometer and path averaged k-R should be discussed. P6L29-32 – earlier you mention that the disdrometer are evenly spread – so is this weighing really important ?

THE k-R ( and not R-K otherwise do not use a and b as in (5 and 6) ) relationships should be given here and the differences between previous studies and IUT discussed

here and not introduced in conclusion. Also here is a good place to discuss point versus path k-R. . ... and your results on this with the 5 disdrometers.

P7 L 19 : the 24h centered window method is not applicable in RT (where you have access only to passed data - - and RT is mentioned in the introduction in the objectives of the work outcomes. . ....

Section 5 :

P7 L 34 what is a 'relatively unambiguous event' ???

P7 L 35 : 'performance . . . for detecting' : this is not done – there is no FAR/Miss study here – only analysis of the rainfall rate itself.

Section 5.2 : As mentioned, this could be enhanced and become the main result section. As suggested by reviewer 1 the analysis in terms of attenuation should be done first and then the retrieved Rain rates can be compared.

One of the major surprise is the discrepancy between the two 38GHz/Hpol links rain retrieval, which is not fully explained by the paper and should be further explored in dB first. -what is the correlation and consistency between the time series of attenuation for the 2 links ? - the variability of the 2 signal in dry/wet periods should be further quantified (variance for instance).

P8 – the analysis should be more objective and vocabulary such as 'visual inspection suggests' L35 ;' the magnitudes are similar' L15 ; 'loss seems almost entirely related to' L28 should be replaced by quantitative indicators.

P9 L1 : 'more spatially heterogeneous and probably convective' – please check and give some indicator of spatial heterogeneity – how is this affecting the path averaged versus punctual k-R on that day ?

P9 L17 to 25 :

The discrepancies between links and between the link and disdro need to be further

understood. What part can be explained by k-R variability ?; what comes from baseline error ? Comparison in DB first (with attenuations derivved from DSD spectra and your Tmatrix code) would help understanding.

Is a possible underestimation of rain rates by the DSD totally eliminated out ? What are the quantitative results of the gauge/DSD comparison for the 3 instruments that are gathered ?

The Conclusion will have to be adjusted when section 5 has been revised.

---

## Referee Comment (RC3) · Anonymous Referee #3 · 15 Jan 2018

Summary:

This manuscript presents results from a comprehensive field experiment studying error sources for rainfall retrieval with microwave links. The paper is well structured and well written, expect for some places where the writing should be made less monotonous. The conclusion are a bit vague, though. However, in my opinion, this is more a short-coming of the writing and less of the experimental setup or the analysis. In summary, this manuscripts provides an important contribution and should undergo a minor revision to be published in AMT.

General comment:

The discussion of the causes, implications and possible mitigation strategies for the

different effects should be more detailed in section 5. In particular the consequences when using data from a large number of operational microwave links from a cell phone network, where no ground truth is available to detect and accurately mitigate the caused errors, should be addressed. It would be important to, at least, estimate the magnitude of the different effects on rainfall retrieval from typical operational microwave link networks.

The title and the abstract do not hold much information about the main goal and findings of this study, the search for explanations of the fluctuations of the received signal level. I recommend that the findings, which are a bit vague, but nevertheless very important for the community of researchers that derive rainfall information from microwave links, are presented clearer already in the abstract.

Specific comments:

Title: The title should reflect the actual topic, investigation of the microwave links errors, a bit more.

page1, Line 18: With all the "and"s this sentence is a bit hard to understand

Page1, Line 25: Why not start a new paragraph here (instead of one sentence before) for the part of the abstract which summarizes the results.

Page1, Line 27: I would not call "temperature" an "attenuating phenomena". As you show in the manuscript, it can have a big effect on the RSL, but not by adding attenuation. It is more likely to be bias from the electronics. Maybe you could reformulate here.

Page1, Line 28: The summary of the conclusions should be more detailed. What is the order of magnitude of the different error sources, etc?

Page 1, Line 36: Instead of "regional" precipitation distribution, writing "local" or just "spatial" fits better here.

Page 1, Line 38: Since the height of the radar observation above ground is very close to the ground near the radar, and can be a lot higher than 1000 meters far from the radar, I would not write "roughly 1000 meters" but maybe mention that it can be more than 1000 meters far from the radar or in complex terrain

Page 2, Line 1: "arsenal" sounds a bit colloquial.

Page 2, Line 10: Do you mean "back then" instead of "since then"

Page 2, Line 13: A bit monotonous: "This. . . This. . . This.."

Page 2, Line 25: Add a comma after "Therefore. . ."

Page 2, Line 25: Is "research. . . into. . ." correct english?

Page 2, Line 25: When speaking about "microphysical aspects" of the "retrieval algorithm" I would think more towards the em-wave scattering of individual drops and not the error sources you are investigating here. Hence, I feel the term "microphysical" is misleading here.

Page 3, Line 31: Add comma after "on the other"

Page 4, Line 24: Is the Nokia link working in both directions? If yes, what is the frequency difference?

Page 4, Line 32: Is the difference of only 176 MHz between the Nokia and the dual-pol RAL link really enough to make sure they do not interfere? To be more precise, do you know the band-pass filtering characteristics of both systems?

Question regarding the systems: Multipath effects can also cause large fluctuations in the received signal level. This effect will be different for different propagation settings, i.e. different frequencies and different antennas. What is the antenna size, beam width and gain for the used systems? Maybe a table with the details of the systems would be good.

Page 4, Line 35: "This provides for comparison in the case of,…". Is the term, "to provide for comparison" correct English?

Page5, Line 7: Since you did not monitor the TX power, how about (temperature) drifts of the transmitter?

Page 5, Line 11: Is "in this way" correct english?

Page 6, line 27: You use temperature observations for deriving the terminal velocity (as stated in section 4.1.1), but here you use a constant temperature of 15 degree Celsius. Why? In particular for 38 GHz, temperature difference e.g. between summer and winter, will impact the extinction cross section and hence the k-R relation.

Page 7, Line 4: You should discuss how do the derived k-R parameters compare to the ones from the literature here.

Page 7, Line 6: It is a bit misleading that you write that you are "closely following" Overeem et al., but some sentences later write that you use a completely different way (which is fine for this experiment) to detect rain events.

Page 7, Line 13: "would not be relevant here", maybe better write "is not applicable here"

Page 8, Line 10: Are you sure this is "background noise", hence stemming from the electronics? It could also stem from propagation differences of the systems, e.g. because of different beam widths or slightly different alignments. Both of these could result in different propagation conditions resulting from diffraction/refraction from the ground (buildings, trees, etc.).

Page 9, Line 1: Add a "a" after "probably"

General question: Doesn't the additive bias mainly stem from the very simple baseline determination?

General question: What is the correlation and bias of the disdrometer and the gauge

next to it?

Page 10, Line 4: The precipitation intensities should not only be "taken with a grain of salt". Their absolute values are, as you explain a little later, completely unusable. Maybe the dynamics indicate a little the dynamics of the precipitation event. But the problem most likely is that your 30-second disdrometer aggregations are too short and only contain a very small number of drops. Hence there is a lot of sharp isolated peaks which might probably stem from individual large snowflakes the disdrometer detected during on 30-second period.

Page 10, line 16: Maybe, if available, you could add possible explanations for this effect if, as you write, the snow deposit alone cannot explain it.

Page 11, line 15: It is not clear from the text whether the periods with high humidity are also removed for the calculation of correlation including rainy periods.

Page 11, line 15: "temperature dependence . . . is still roughly consistent. . .". First of all, the term "still roughly consistent" is a bit vague and should be rephrased. However, judging from table 1, there is a clear decrease of the correlation if rainy periods are included and a further decrease when only considering rainy period. I think, this should be reflected in the text. Nevertheless, the correlation of, e.g. -0.4 for rainy periods only, is surprisingly high.

Page 11, line 26: Does dew really build up a thin layer on the antenna or does it also form small droplets, as shown for the spraying of the antennas?

Page 11, line 41: Fog cannot generate an attenuation of 3 dB for such short links at frequencies of 38 GHz and below (please check the references you cited). Hence, I do not understand why the you consider fog as a possible source in this sentence.

Page 13, line 19: ". . .no lingering attenuation,. . ." This comma should be moved after "..in both cases. . .".

Page 14, line 2: The comparison of the different parameters would fit better in section

4.1.3 where the actual analysis is explained. In the conclusion section I would not expect the presentation of new results or data.

Page 14, line 43: Will the data of the experiment be made available after publication?

Fig 5: "RAL 38 V" appears two times in the legend. Colors of RAL 38H and RAL 26 change between plot "a" and "b"

Fig 7: Why not use the same color for the different microwave links as in the time series plots, e.g. Fig 6.
* * *

---

## Author Comment (AC1) · 16 Feb 2018

REVIEWER: General comment: The manuscript describes dedicated experiment designed to investigate different phenomena influencing rainfall retrieval from microwave links. Several microwave links were installed over same path and equipped with time lapsed cameras shooting antenna surfaces and the link path. In addition, array of disdrometers completed with rain gauges were placed along the link path. Finally, additional observations form nearby weather station such as temperature, humidity or wind speed were used to interpret phenomena occurring during the measurement campaign. The manuscript goal is to provide comprehensive overview of different phenomena causing attenuation of microwave links and evaluate their relevance for rainfall intensity retrieval, specifically to the rainfall retrieval algorithm as suggested by

[Figure]

Overeem et al. (2011 and 2016). The first goal is scientifically relevant as i) it might improve understanding of uncertainties affecting microwave link rainfall retrieval and ii) description of attenuation patterns from other phenomena than rainfall is crucial for improving baseline separation algorithms. The presented experimental setup is very well suited to provide reliable dataset to reach this goal. The second goal is bit too specific to the selected processing algorithms (Overeem et al. 2011 and 2016).

RESPONSE: We thank the reviewer for the positive assessment of our paper. We acknowledge that the second stated goal is too specific and this does not in fact reflect our actual intentions. The mention of the algorithms of Overeem et al (2011 and 2016) is intended merely as an example of possible integration in existing retrieval schemes and not as a goal for this paper. The text as written in P2L27-29 does not properly reflect this and we will revise it.

REVIEWER: The manuscript focuses on describing different phenomena causing link attenuationon several selected events. Overall statistical evaluation is mostly not provided which hinders quantitative assessment of the influence of these phenomena on microwave link rainfall retrieval. Results are often presented qualitatively in subjective manner (e.g. 'link is remarkably stable') even in cases where it could be easily described quantitatively, for more details see specific comments.

RESPONSE: We agree with the reviewer that many specific instances can be easily described more quantitatively and we will do so in the revised manuscript. Please see our resposes to the specific comments for more details.

REVIEWER: Authors should distinguish in the whole result section more properly if the attenuation occurs along the link path or if it is rather related to hardware of microwave link radio units/antennas. The ambiguous cases should be then properly discussed and possibly confronted with radio wave propagation theory or results of other studies.

RESPONSE: We will add clarification to the different parts of the results section where applicable. The ambiguous cases are mostly illustrative and a more thorough analysis

using radio wave propagation theory is beyond the scope of the current paper, but will be part of future work. Moreover, we are not aware of similar ambiguous cases described in the scientific literature.

REVIEWER: The manuscript is well structured, however, stylistics might be still improved, e.g. paragraphs in the result section could be more concise and fluent.

RESPONSE: We will carefully re-read the manuscript and apply modifications where applicable.

REVIEWER: Specific comments:

P7L28: Results and discussion section: The results of microwave links are in the text mostly presented in mm/h although figures show also dBs. I strongly recommend to present the results also in dBs and compare them with theoretical rain induced attenuation from disdrometer data (eq. 3). The main reasons are these i) the uncertainties arising from imperfect separation of rain-induced attenuation are mixed with uncertainties arising from rainfall-attenuation powerlaw model, i.e. variability of $\alpha$ and $\beta$ parameters (Tab. 2) during different rainfall events and uncertainties due to path-integration of attenuation and nonlinearity of power-law model. This hinders interpretation of results. ii) Substantial part of link attenuation unexplained by raindrops are hardware related errors (e.g. due to wet antenna or quantization noise). Such uncertainties expressed in mm/h apply only to links of the same lengths as in the experiment. iii) Most of the literature concerning microwave link propagation and different phenomena influencing radio wave attenuation (including wet antenna attenuation) express results in dBs.

RESPONSE: We completely agree with the reviewer on this point. We will add an extra panel to figures 5, 6, 8, 9, and 13, showing attenuation in dBs including the disdrometer-derived theoretical attenuations.

REVIEWER: P8L5: It is stated here that in the presented event there are 'no attenuation-inducing influences other than rain', however, this is inexact as the radio

waves are during this event for sure attenuated e.g. by atmospheric gases, there is a free space loss, etc.

RESPONSE: This is meant as "no attenuating phenomena contributing to the dynamics of the signal". We will clarify this in the text.

REVIEWER: P8L33: There are certainly various attenuating phenomena (see comment P8L5) influencing link attenuation, and drop down in the RAL link signal level has probably some (uknown?) reason.

RESPONSE: We agree. We will alter the text on this point.

REVIEWER: P9L4: 'remarkably stable' or 'uncertain baseline' is very subjective description. Pleasequantify.

RESPONSE: Agreed. We will provide numbers in the revised version.

REVIEWER: P9L17-20: The causes of outliers and overestimation discussed in these lines are speculative. The experimental design should enable investigate unexpected behavior of links much more specifically thanks to reliable ground truth, cameras, etc. For example, it is stated here that 'overestimation and outliers could be attributed to attenuating phenomena ... erroneously processed as rain in the basic algorithm'. It should be, however, possible to check against disdrometer data if the errors are due to the processing algorithm. Similarly, errors introduced by k-R model can be estimated and it should be verified if they can explain underestimation.

RESPONSE: We have added to this response a new figure illustrating the relation between link attenuation and disdrometer-derived theoretical attenuation at the relevant frequencies. We also added an updated version of figure 7. These pictures show very similar results. Therefore, the R-k power law model introduces very little additional error. This is further supported by the high goodness-of-fit found for the R-k model itself. We will further clarify this in the text.

REVIEWER: P10L22-24: Please quantify the magnitude of oscillations.

[Figure]

RESPONSE: Agreed. We will quantify this magnitude.

REVIEWER: P10L33: Is the 90 % humidity threshold selected arbitrary, based on radiowave propagation theories, or estimated by regression itself? Please indicate.

RESPONSE: The reason for this is made clear in P11L8-11 when talking about dew. We will re-arrange the text so that this is clear when the 90% threshold is introduced.

REVIEWER: P11L13-15: The statement that 'the temperature dependence of the Nokia link is drowned out in the noise' is speculative as you cannot prove there is a temperature dependency if it is 'drowned out in the noise'. If you can prove it (at reasonable confidence level) it is then not 'drowned out in the noise'.

RESPONSE: We acknowledge that this phrasing is sloppy. We have rephrased: "There is no evidence of a temperature dependence of the Nokia link here, even though one would expect it based on the findings from 14-24 April."

REVIEWER: P11L19: Please indicate in the text the duration of antenna wetting and drying quantitatively. The figures depict too long period to distinguish if the processes take place only few minutes, tens of minutes or few hours.

RESPONSE: The timescale of these events is in the order of hours. We will add this information to the text.

REVIEWER: P11L31-32: Please describe more precisely what is meant with 'quite different pattern'. Different range, variability, autocorrelation structure, ...?

RESPONSE: We refer here to the autocorrelation structure. We will clarify this in the text and provide more detail.

REVIEWER: P13L11-12: What is meant with 'any other atmospheric phenomena'? Furthermore, the following text relates the attenuation to the humidity which is an atmospheric phenomena. The whole meaning of this sentence is, therefore, unclear.

RESPONSE: restated: "by any one atmospheric phenomenon as described in the previous sections"

REVIEWER: P13L15-22: The antenna drying times might be very much influenced also by other environmental variables such as wind or sun radiation. Could e.g. wind which is also displayed in the figs 16 and 17 explain part of the uncertainty in drying duration? Is there any reason why humidity is included in the quantitative analyses and not the wind?

RESPONSE: While an interesting topic, this would go beyond the scope of this paper. We will mention in the text that these phenoma could indeed influence the antenna drying times and that this is subject of future work.

REVIEWER: P14L20: A robust evidence that the link response to the additive and multiplicative bias is consistent over different events has not been provided in the previous text. Why don't you e.g. quantify both additive and multiplicative bias for each event and link and provide information about range and variability of both types of biases?

RESPONSE: Our statement refers to P9L5-14 and fig. 7. We feel that this provides enough evidence to make this statement. Computing these biases for each single event in the data set is beyond the scope of this paper.

REVIEWER: Figures: There is a wrong legend in the panel (a) of the figures 5, 6, 8, 9, 10, 12, 15, 16 and 17 as RAL 38 V is assigned to both blue and green lines. It seems to be that green line belongs to the RAL 38 H and the orange one to the RAL 26 V, i.e. same coding as in the panel (b)

RESPONSE: We thank the reviewer for catching this error! We will correct this.
* * *
**Fig. 1.** scatterplots of link-derived rainfall intensities versus disdrometer-derived rainfall intensities

**Fig. 2.** scatterplots of link attenuation versus disdrometer derived attenuation

---

## Author Comment (AC2) · 16 Feb 2018

REVIEWER: The authors have put together an experimental campaign in order to explore in depth some aspects of rainfall measurement from microwave links attenuation and understand better the uncertainties, focusing an a relatively short links (2 km) and medium high frequencies (26/38 GHz). The experimental set up is impressive and thorough, with 2 microwaves links –one operating at 38 Ghz and the other one dual frequency 38/26 GHz and dual polarization- sharing the same (2 km long) path ; 5 disdrometers in order to analyze rainfall intensity and microphysical variability along the path and at both ends, and an additional rain gauge ; a near IR scintillometer, cameras and a met station complement the set up and provide additional information on visibility and other atmospheric variables that might influence or help understanding the MWlinks signal fluctuations. This is an ideal setting to analyze and quantify – at least for the 2 frequencies and the path length that are available here – some of the uncertainties in MWlinks based Quantitative Precipitation Estimation : -variability and time/space scale issues in the k-R relationship -wet antenna attenuation -baseline derivation uncertainty and non-rain induced fluctuations of the signal

-Additionally the MWlink power level is sampled at 20 Hz and with a small quantization error - which could be used to investigate errors due to coarse quantization and to the subsampling of the signal typically provided by Commercial MWLinks network monitoring systems (only providing max and min power every 15 minutes is common).

-Also the dual-frequency and dual-pol capability, together with the 5 disdrometers will allow to go further than the simple k-R based retrieval.

First of all I would like to congratulate the authors for the experiment they have put together and acknowledge the amount of work and time which will be necessary to fully exploit such a data set !

RESPONSE: We thank the reviewer for the appreciation of our experiment and paper.

REVIEWER: The main objective of the present paper is to present the experimental setting itself and some preliminary results which illustrate -in a rather qualitative manner- some of the issues that could be further explored with the data set : the discrepancy between the rainfall retrieved by the links and the path average rainfall retrieved by the 5 disdrometers, illustrated with 3 rainfall events ; some illustration of measurement during mixed precipitation ; effects of temperature on the signal fluctuations; wet antenna attenuation and its sensitivity to the type of radom material ; effect of dew and fog ; effect of clutter. Altogether an interesting catalog that illustrates the complexity of mwlink based retrieval of precipitation is provided in section 5. However, the reader stands a bit frustrated by a somehow QUALITATIVE OVERVIEW OF VARIOUS CAUSES OF MW SIGNAL FLUCTUATIONS, WITH A LACK OF STATISTICAL AND QUANTITATIVE ASSESSMENT OF THEIR IMPACT ON I) DETECT-

ING/QUANTIFYING ATTENUATION DUE TO RAIN AND II) RETRIEVING RAIN RATE.

RESPONSE: As the reviewer acknowledges, the point of this paper was to provide a somewhat qualitative overview of the issues which could be explored with this dataset. An in-depth analysis of any one of these issues would merit it's own separate treatment and is beyond the scope of this paper. We do intend to explore several of these issues further and we encourage others to do so as well once we have published the accompanying dataset. However, as we have also admitted in our response to referee #1, some of our statements in the results section are needlessly qualitative and could easily be made more quantitative. We will remedy this in a revised manuscript.

REVIEWER: I would suggest that Sub-section 5.2, which has the most quantitative results and focus on the main objective of the MWlinks exploitation, i.e. rain retrieval, become a full section and be improved with some more quantitative results.

The other sub-sections in 5 should be lightened (5.3 and Fig9 which is essentially qualitative can be suppressed) and more focused on explaining some of the discrepancies observed in 5.2.

RESPONSE: We believe that following the recommendation in this comment would not serve to improve the manuscript as it would radically alter the focus of the paper. We will hence keep the structure as it was. However, we will add more quantitative results wherever possible, as also indicated in our response to reviewer #1.

REVIEWER: The text itself needs revising ; some expressions or comments are more subjective than scientific – and the authors sometimes overgeneralize their statements e.g. P4 L1 'the power law in the literature are ALL derived at point scale ' P 13 L12 'it is important to take into account that there will always be unexplained anomalies' P2 l 39 :' a relatively straightforward algorithm' P3 L31 'the relations ..... closely resemble power law' etc... see more below.

RESPONSE: We will revise the overgeneralized statements P4 L1 and P 13 L12. However, we do not believe that statements such as P2 L39 and P3 L31 are problematic. We will adapt the manuscript where applicable to make the wording less subjective.

REVIEWER: DETAILed SUGGESTIONS : Title/Introduction :

- The stress on urban in the title is misleading – the paper does not focus on an urban problem or urban hydromet scales specially. A tile like 'A multi-instrument microwave link measurement campaign' would be better. - The introduction also stresses a lot on urban scales which is not that relevant since a single link and not a dense network are studied here.

RESPONSE: The frequencies employed here are frequencies that are often used in operational networks in urban areas. Furthermore, although the experiment features a single link, the context for these research questions comes from the use of urban link networks. However, we acknowledge that the experiment itself is not necessarily only applicable to urban applications and we will adapt the title to reflect that.

REVIEWER: - P1 L33-L35 confusion between the space/time resolution of a single gauge and the problem of gauge network density versus scale of phenomenon.

RESPONSE: This text will be revised to make this distinction clearer.

REVIEWER: - P2l8 : modern radar also used propagation variables such as Kdp and not just Z....

RESPONSE: "radar" is now changed in the text to "traditional radar". Note that Kdp can only be used for rainfall estimation at high rain rates, so that even with dual-pol radar a relation between Z and R is needed for the lower rain rates.

REVIEWER: -P2 L25 : 'therefore further research... ... microphysical aspects' – Sentence not clear + microphysics is not really the focus of this paper....

RESPONSE: We will modify the text to read "Therefore, further research is needed regarding the physical aspects of the attenuation measurements themselves.".

REVIEWER: - P2 L 27 : relevance of urban? ' in order to help fine tune the existing retrieval algorithm' - The sentence is clumsy and there is nothing about tuning the algorithm in the presented work anyway..... L 29: simulating links from radar data is not at all relevant to what is proposed here and to the local scale ( one single 2 km path) studied.

RESPONSE: P2L27-29 will be completely revised in the updated manuscript, and the sentence related to "simulated links" will be removed.

REVIEWER: Section 2 – P3 L 29 ' the relations .... Resemble power laws' – Paragraph to be revised – false or approximative statements.

RESPONSE: It is not clear to us what the referee means here. We see no false statements here. We will revise the wording of this paragraph to make it clearer.

REVIEWER: k-R relationship discussion : the discussion on k-R is spread in different parts of the manuscript with no consistency . The paragraph starting on P3 L35 is rather confused. There seem to be a confusion between DSD/rain type variability between rainfall events and k-R variability as a function of the scale considered (point versus path...). The concept of 'control volum' is unexplained and unclear. P3 Eq(6) insists on the problem of linearity of the relationship and the problem of point versus path average k-R relationship - but the fact that in this work the path average k-R relationship is effectively derived thanks to 5 disdrometers in not mentioned. ...Subsection 4.1.2-4.1.3 should be merged and with a more explicit title like 'deriving path averaged k-R relationship'.

RESPONSE: We do not use path-averaged rain rate and attenuation to derive an R-k relationship, but the point scale data from all disdrometers are employed. This would not make much of a difference anyway. The importance of the near-linearity of the R-k relationship is related to the variability of raindrop concentration and size distribution along the path and not directly to variability between different events or as a function of the scale considered. Using a path-averaged relationship would not solve the ambiguity

unless one derives a new R-k relationship for every timestep for this particular path and rain field. This would defeat the point of doing a microwave retrieval in the first place! We admit that the phrasing used in P4L1 is misleading and we will revise it.

REVIEWER: Also note that k=aRËEₑb is used in (5 and 6) while a and b in Table 2 are for R=a kËEₑb most confusing

RESPONSE: Equation 5 and 6 are incorrect. Thank you for noting this. We will correct this.

REVIEWER: Section 3 : P4 L25 – please give the same precision for the frequency of the Nokia and RAL links.

RESPONSE: We will modify this.

REVIEWER: L24 : 'representative of THE link systems that would be used ....' to be rephrased carefully – not all CMLs are Nokia and you dont use the NOKIA sampling/digitalization....

RESPONSE: "representative" is changed in the text to "a typical example".

REVIEWER: P4 L 31 : 'roughly' – unprecise L32-33 give the exact frequencies.

RESPONSE: See our response above.

REVIEWER: Section 4 4.1.2/4.1.3 – merge and improve. It seems that you are deriving a path-averaged k-R relationship based on weight values of both k and R derived from the 5 disdrometers with 30s long DSD spectra. Is this the case ? not very clear from the text. This very important point should stressed : most studies do not have access to the path average k-R and have to infer it from ponctual k-R and assumptions on rainfall variability . The differences between the single disdrometer and path averaged k-R should be discussed.

RESPONSE: This is not the case and that fact is mentioned in the text : P6L40. We will add an extra clarification to P6L29.

REVIEWER: P6L29-32 – earlier you mention that the disdrometer are evenly spread – so is this weighing really important ?

RESPONSE: They are not evenly spread so this is important. This can be seen in Fig 1a. They were only as evenly spread as the limitations of the underlying terrain allowed (mainly access to large flat rooftops). This is maybe not clear from the phrasing in P5L14 so we will add a clarification here.

REVIEWER: THE k-R ( and not R-K otherwise do not use a and b as in (5 and 6) ) relationships should be given here and the differences between previous studies and IUT discussed here and not introduced in conclusion. Also here is a good place to discuss point versus path k-R.... and your results on this with the 5 disdrometers.

RESPONSE: We welcome this suggestion. We moved some of the discussion from section 6 to here.

REVIEWER: P7 L 19 : the 24h centered window method is not applicable in RT (where you have access only to passed data - - and RT is mentioned in the introduction in the objectives of the work outcomes.....

RESPONSE: We do not believe this is relevant. We never suggest that the methods used in this paper are directly applicable to operational settings.

REVIEWER: Section 5 : P7 L 34 what is a 'relatively unambiguous event' ???

RESPONSE: An event that can be related a single type of attenuating phenomenon, such as rain or dew formation on the antennas. In contrast to the compound phenomena in section 5.8. We will rephrase the text to make this clearer.

REVIEWER: P7 L 35 : 'performance ... for detecting' : this is not done – there is no FAR/Miss study here – only analysis of the rainfall rate itself.

RESPONSE: changed "detecting" to "measuring"

REVIEWER: Section 5.2 : As mentioned, this could be enhanced and become the main

result section. As suggested by reviewer 1 the analysis in terms of attenuation should be done first and then the retrieved Rain rates can be compared. One of the major surprise is the discrepancy between the two 38GHz/Hpol links rain retrieval, which is not fully explained by the paper and should be further explored in dB first. -what is the correlation and consistency between the time series of attenuation for the 2 links ? - the variability of the 2 signal in dry/wet periods should be further quantified (variance for instance).

RESPONSE: As we mentioned above, making this section the main results section of this paper would greatly alter the focus of this paper, and we do not intend to do that. In our responses to reviewer #1, we have indicated that we will add comparisons in terms of specific attenuation. We have also indicated in our responses to reviewer #1 that we will quantify the signal fluctuations. We will also add an additional paragraph detailing the discrepancies between the links including their correlation.

REVIEWER: P8 – the analysis should be more objective and vocabulary such as 'visual inspection suggests' L35 ;' the magnitudes are similar' L15 ; 'loss seems almost entirely related to' L28 should be replaced by quantitative indicators.

RESPONSE: We will revise these phrases.

REVIEWER: P9 L1 : 'more spatially heterogeneous and probably convective' – please check and give some indicator of spatial heterogeneity – how is this affecting the path averaged versus punctual k-R on that day ?

RESPONSE: Spatial heterogeneity can be indicated by the spatial coefficient of variation. We will add these numbers to the text. The high rainfall variability in time and space and the high rainfall rates are all indicative of convective rainfall. It has been shown (e.g. Berne and Uijlenhoet, GRL, 2007, Leijnse et al., JHM, 2010) that the effect of spatial variability on the k-R relation is limited at the frequencies under consideration.

REVIEWER: P9 L17 to 25 : The discrepancies between links and between the link and disdro need to be further understood. What part can be explained by k-R variability ?; what comes from baseline error ? Comparison in DB first (with attenuations dervived from DSD spectra and your Tmatrix code) would help understanding.

RESPONSE: See our response to comment P9L17-20 by reviewer #1.

REVIEWER: Is a possible underestimation of rain rates by the DSD totally eliminated out ? What are the quantitative results of the gauge/DSD comparison for the 3 instruments that are gathered ?

RESPONSE: This is beyond the scope of this paper.

REVIEWER: The Conclusion will have to be adjusted when section 5 has been revised.

RESPONSE: We will modify conclusions based on the modifications made in this paper. Note that the modifications that we will make are less than what reviewer #2 suggests. Hence the necessary modifications to the conclusions will be minor.

---

## Author Comment (AC3) · 16 Feb 2018

REVIEWER: Summary: This manuscript presents results from a comprehensive field experiment studying error sources for rainfall retrieval with microwave links. The paper is well structured and well written, expect for some places where the writing should be made less monotonous. The conclusion are a bit vague, though. However, in my opinion, this is more a shortcoming of the writing and less of the experimental setup or the analysis. In summary, this manuscripts provides an important contribution and should undergo a minor revision to be published in AMT.

RESPONSE: We thank the referee for the appreciative words. We will reformulate some of the vagueness in the text. See the specific comments for details.

REVIEWER: General comment: The discussion of the causes, implications and possible mitigation strategies for the different effects should be more detailed in section 5. In particular the consequences when using data from a large number of operational microwave links from a cell phone network, where no ground truth is available to detect and accurately mitigate the caused errors, should be addressed. It would be important to, at least, estimate the magnitude of the different effects on rainfall retrieval from typical operational microwave link networks.

RESPONSE: Extension of the analysis to link networks is outside the scope of this work. However, we will provide a more thorough discussion of the causes, implications, and possible mitigation strategies for the different effects that we encounter in our data. This includes a discussion of the magnitude of the effects of the different phenomena.

REVIEWER: The title and the abstract do not hold much information about the main goal and findings of this study, the search for explanations of the fluctuations of the received signal level. I recommend that the findings, which are a bit vague, but never-theless very important for the community of researchers that derive rainfall information from microwave links, are presented clearer already in the abstract.

RESPONSE: The Abstract will be adapted to include more of the findings.

REVIEWER: Specific comments: Title: The title should reflect the actual topic, investigation of the microwave links errors, a bit more.

RESPONSE: The title will be replaced by one more appropriate to the contents of the paper.

REVIEWER: page1, Line 18: With all the "and"s this sentence is a bit hard to understand

RESPONSE: This sentence will be modified to improve its readability.

REVIEWER: Page1, Line 25: Why not start a new paragraph here (instead of one sentence before) for the part of the abstract which summarizes the results.

RESPONSE: We will implement this suggestion.

REVIEWER: Page1, Line 27: I would not call "temperature" an "attenuating phenomena". As you show in the manuscript, it can have a big effect on the RSL, but not by adding attenuation. It is more likely to be bias from the electronics. Maybe you could reformulate here.

RESPONSE: "attenuating phenomena" will be changed to "phenomena affecting received signal level".

REVIEWER: Page1, Line 28: The summary of the conclusions should be more detailed. What is the order of magnitude of the different error sources, etc?

RESPONSE: We agree with the reviewer that adding details about the magnitude of the effects of the different error sources will improve the paper. We will hence add general quantitative results.

REVIEWER: Page 1, Line 36: Instead of "regional" precipitation distribution, writing "local" or just "spatial" fits better here.

RESPONSE: Thank you for the suggestion. We will use "spatial".

REVIEWER: Page 1, Line 38: Since the height of the radar observation above ground is very close to the ground near the radar, and can be a lot higher than 1000 meters far from the radar, I would not write "roughly 1000 meters" but maybe mention that it can be more than 1000 meters far from the radar or in complex terrain

RESPONSE: We will change the text following the recommendation of the referee.

REVIEWER: Page 2, Line 1: "arsenal" sounds a bit colloquial.

RESPONSE: We will replace "arsenal" by "range"

REVIEWER: Page 2, Line 10: Do you mean "back then" instead of "since then"

RESPONSE: We do mean "since then", but we agree that the way it is used in this

sentence is confusing. We will rewrite the sentence to become: "Despite these advantages, microwave links have not been deployed at a large scale for the purpose of precipitation monitoring, for the cost of setting up such a network would still have been quite severe.".

REVIEWER: Page 2, Line 13: A bit monotonous: "This ... This ... This..."

RESPONSE: We will some variation in the wording.

REVIEWER: Page 2, Line 25: Add a comma after "Therefore ..."

RESPONSE: We will add a comma.

REVIEWER: Page 2, Line 25: Is "research ... into ... " correct english?

RESPONSE: We will rewrite this to become: "Therefore, further research is needed regarding the physical aspects of the attenuation measurements themselves.".

REVIEWER: Page 2, Line 25: When speaking about "microphysical aspects" of the "retrieval algorithm" I would think more towards the em-wave scattering of individual drops and not the error sources you are investigating here. Hence, I feel the term "microphysical" is misleading here.

RESPONSE: See our response to the comment about this to reviewer #2. We will modify the text to read "Therefore, further research is needed regarding the physical aspects of the attenuation measurements themselves.".

REVIEWER: Page 3, Line 31: Add comma after "on the other"

RESPONSE: We will add a comma here.

REVIEWER: Page 4, Line 24: Is the Nokia link working in both directions? If yes, what is the frequency difference?

RESPONSE: The Nokia link is bidirectional, but only the received signal level at one end was recorded. The frequency of the reverse carrier wave is 39.436250 GHz.

REVIEWER: Page 4, Line 32: Is the difference of only 176 MHz between the Nokia and the dual-pol RAL link really enough to make sure they do not interfere? To be more precise, do you know the band-pass filtering characteristics of both systems?

RESPONSE: The bandwidth of the RAL receiver is 4 KHz, and the bandwidth of the Nokia receiver is 0.9 MHz. The bandwidth of the RAL transmitter is « 1 KHz and the bandwidth of the Nokia transmitter is 3.5 MHz. The difference of 176 MHz should therefore be enough to avoid interference. We will add additional information to the description of the links in section 3.2.

REVIEWER: Question regarding the systems: Multipath effects can also cause large fluctuations in the received signal level. This effect will be different for different propagation settings, i.e. different frequencies and different antennas. What is the antenna size, beam width and gain for the used systems? Maybe a table with the details of the systems would be good.

RESPONSE: We will provide additional information on the antenna characteristics.

REVIEWER: Page 4, Line 35: "This provides for comparison in the case of, ... ". Is the term, "to provide for comparison" correct English?

RESPONSE: We believe it is. However, understand that the sentence may be difficult to read. We will alter this sentence to improve its readability: "This provides information about, for example, fog and other visibility-affecting phenomena.".

REVIEWER: Page5, Line 7: Since you did not monitor the TX power, how about (temperature) drifts of the transmitter?

RESPONSE: We do not make a distinction between temperature dependencies in the transmitter or the receiver.

REVIEWER: Page 5, Line 11: Is "in this way" correct english?

RESPONSE: We will modify this part of the sentence to become: "This information can

be used to, for example, identify solid precipitation in the...".

REVIEWER: Page 6, line 27: You use temperature observations for deriving the terminal velocity (as stated in section 4.1.1), but here you use a constant temperature of 15 degree Celsius. Why? In particular for 38 GHz, temperature difference e.g. between summer and winter, will impact the extinction cross section and hence the k-R relation.

RESPONSE: This was done due to pragmatic reasons. However, the temperature dependence of the extinction cross section is very slight within the used temperature range and will not effect the R-K relation to a significant degree. See e.g. Olsen et al. (1978).

REVIEWER: Page 7, Line 4: You should discuss how do the derived k-R parameters compare to the ones from the literature here.

RESPONSE: We have moved the relevant parts of section 6 to section 4.1.3, so that this section now also includes a comparison with values from the literature.

REVIEWER: Page 7, Line 6: It is a bit misleading that you write that you are "closely following" Overeem et al., but some sentences later write that you use a completely different way (which is fine for this experiment) to detect rain events.

RESPONSE: This sentence is indeed misleading and an unfortunate remnant of earlier drafts. We will remove it.

REVIEWER: Page 7, Line 13: "would not be relevant here", maybe better write "is not applicable here"

RESPONSE: Agreed. We will rephrase this according to the reviewer's suggestion.

REVIEWER: Page 8, Line 10: Are you sure this is "background noise", hence stemming from the electronics? It could also stem from propagation differences of the systems, e.g. because of different beam widths or slightly different alignments. Both of these could result in different propagation conditions resulting from diffraction/refraction from

the ground (buildings, trees, etc.).

RESPONSE: By using the term "background noise" we did not wish to imply that the source is with the electronics, but rather that the source is unknown and not related to the quantity of interest. Moreover, we are not talking about the "dry" received power level per se, but rather the variability within this power level for a single receiver as represented by the 5th and 95th percentile over the moving window. We will clarify what we mean by the "background noise level".

REVIEWER: Page 9, Line 1: Add a "a" after "probably"

RESPONSE: That does not seem right to us, as we have an "a" between "is" and "more".

REVIEWER: General question: Doesn't the additive bias mainly stem from the very simple baseline determination?

RESPONSE: Yes. We will expand upon this in the revised text.

REVIEWER: General question: What is the correlation and bias of the disdrometer and the gauge next to it?

RESPONSE: We will add this information to the manuscript.

REVIEWER: Page 10, Line 4: The precipitation intensities should not only be "taken with a grain of salt". Their absolute values are, as you explain a little later, completely unusable. Maybe the dynamics indicate a little the dynamics of the precipitation event. But the problem most likely is that your 30-second disdrometer aggregations are too short and only contain a very small number of drops. Hence there is a lot of sharp isolated peaks which might probably stem from individual large snowflakes the disdrometer detected during on 30-second period.

RESPONSE: We agree with the reviewer that the absolute values are incorrect. However, the dynamics of the intensity and the precipitation type are useful information,

which is why we stated that they should be taken with "a grain of salt". We will rephrase this subjective sentence to clarify this.

REVIEWER: Page 10, line 16: Maybe, if available, you could add possible explanations for this effect if, as you write, the snow deposit alone cannot explain it.

RESPONSE: One possible explanantion is antenna wetting due to the partial melting of the snow deposits on top of the antenna cover. However, at this point that is no more than speculation and we would need more research to confirm it.

REVIEWER: Page 11, line 15: It is not clear from the text whether the periods with high humidity are also removed for the calculation of correlation including rainy periods.

RESPONSE: We have not removed periods with high humidity for the computation of correlation coefficients in rain. We will clarify this in the text.

REVIEWER: Page 11, line 15: "temperature dependence ... is still roughly consistent ...". First of all, the term "still roughly consistent" is a bit vague and should be rephrased. However, judging from table 1, there is a clear decrease of the correlation if rainy periods are included and a further decrease when only considering rainy period. I think, this should be reflected in the text. Nevertheless, the correlation of, e.g. -0.4 for rainy periods only, is surprisingly high.

RESPONSE: We will revise the text accordingly.

REVIEWER: Page 11, line 26: Does dew really build up a thin layer on the antenna or does it also form small droplets, as shown for the spraying of the antennas?

RESPONSE: The effect is the same as for the artificial spraying. So, a nearly uniform layer on the Nokia and large drops on the RAL links. This can be observed in the time lapse camera footage.

REVIEWER: Page 11, line 41: Fog cannot generate an attenuation of 3 dB for such short links at frequencies of 38 GHz and below (please check the references you cited).

Hence, I do not understand why the you consider fog as a possible source in this sentence.

RESPONSE: We agree with the reviewer that fog cannot generate an attenuation of 3dB for these links. However, part of the attenuation (up to 1.5 dB) can be caused by fog on the path, so this is why we do mention this here. We agree that the sentence can benefit from rewording to clarify what we mean: "However, both wetting of the antennas and the attenuation by the fog droplets themselves can contribute to this attenuation, and it is difficult to estimate their respective relative contributions to the total attenuation."

REVIEWER: Page 13, line 19: "... no lingering attenuation, ..." This comma should be moved after "..in both cases ...".

RESPONSE: We will move this comma.

REVIEWER: Page 14, line 2: The comparison of the different parameters would fit better in section 4.1.3 where the actual analysis is explained. In the conclusion section I would not expect the presentation of new results or data.

RESPONSE: This section will be moved.

REVIEWER: Page 14, line 43: Will the data of the experiment be made available after publication?

RESPONSE: Yes! We believe is is very important that others can also use this dataset. The raw data will be published as soon as we have made final quality control checks. In the mean time, the data will be available upon request from the corresponding author.

REVIEWER: Fig 5: "RAL 38 V" appears two times in the legend. Colors of RAL 38H and RAL 26 change between plot "a" and "b"

RESPONSE: This is unfortunate. The graphs will be fixed.

REVIEWER: Fig 7: Why not use the same color for the different microwave links as in

the time series plots, e.g. Fig 6.

RESPONSE: This is a welcome suggestion. We will do so.

---

## Author Response (AR2)

**Editor Report**

*We have received reviews of your revised AMT manuscript amt-2017-404 ("A measurement campaign to assess sources of error in microwave link rainfall estimation"). Referee 2 is satisfied with your revisions. Referee 1, however, still recommends a somewhat lengthy list of suggested improvements that will increase both the readability and scientific impact of your manuscript. While I recommend carefully addressing all of the points highlighted by Referee 1 in your revised manuscript and response to reviewers, I specifically request that you address the lack of quantitative analyses, improve wording in key sections highlighted by Referee 1, and devise a more holistic conclusion that covers important findings raised in your study that are somewhat overlooked in your concluding remarks.*

RESPONSE:

We thank the editor for the suggestions. We have hopefully addressed all serious wording issues in the second revision. The biggest change in this revision is the conclusions section, which now better reflects all the subtopics addressed in the results section.

**Anonymous Referee #1**

**REVIEWER**:

*The revised manuscript mostly reflects the specific comments raised by the reviewers. The authors took effort to quantify phenomena occurring at experimental microwave link which were before described qualitatively and often in the subjective manner. Nevertheless, there are still some parts which should be clarified and formally better described. Unfortunately, the readability of the whole manuscript, and in particular the results section, which lacks clearness and reasonable conciseness, has not improved. It is often very difficult to follow the reasoning and extract from the paragraphs what it is the message the authors want to present. Some examples and suggestions are provided in the specific comments; nevertheless, especially comments on language shortcomings are not exhaustive. The authors should once more carefully read through the whole manuscript, particularly the result section, and do editing where appropriate. A stylistic and language proofread by an English native speaker might also help. The presentation quality is the main reason why I recommend major revisions and not just minor.*

**RESPONSE:**

We thank the reviewer for recognizing that we revised the manuscript according to the reviewers' comments mostly satisfactorily. We have improved the wording of the results section where appropriate. Apart from the adaptations made in response to the specific comments we have made a few minor corrections and clarifications. However, we do not agree with the reviewer that the manuscript as a whole has severe language shortcomings.

**REVIEWER:**

*Finally, the authors should consider revising conclusion section. Introducing paragraph of the section is devoted to the effect of drop size distribution on power-law fit, however, I do not see any scientifically novel findings in this particular issue. The power-law fitting to DSD is even not a part of results section. In contracts, e.g. the attenuation dependence on the temperature (the whole subsection 5.4 is devoted to this), is not mentioned in the conclusions at all, despite the presented research clearly improves current understanding of baseline variability sources. Similarly, conclusions based on result subsections on solid precipitation (5.3), dew and fog (5.5), Clutter (5.7) and Compound phenomena (5.8) are not provided. I personally also miss more specific recommendations either how to cope with the effects described in the manuscript when estimating rainfall (e.g. how to use redundant information from more CMLs) or what lines of research are still needed to better cope with these effects. It should be noted, that the criticism on the conclusion section has not been raised in previous steps of the review process.*

**RESPONSE:**

We agree with the reviewer that the conclusions as written lack connection with the results. The paragraph on the power-law fitting has been removed and instead several paragraphs have been added drawing conclusions from each of the result subsections in order. The focus of our paper is to present a measurement campaign to address several error sources associated with rainfall estimates from microwave links in cellular communication networks. This includes a preliminary global analysis and several cases highlighting the different phenomena affecting received signal level. Specific recommendations how to cope with (sources of) errors in CML rainfall estimation, i.e. how to improve rainfall retrieval algorithms, are beyond the scope of this paper.

**REVIEWER:**

*Specific comments:*
*P2L37: part of the sentence is missing.*

**RESPONSE:**

This is not true, although we can see how the marked-up text could have been a cause for confusion

 here. The full sentence reads: "Therefore, further research is needed regarding the physical aspects of the attenuation measurements themselves."

**REVIEWER:**

*P3-4 Theoretical background section: This section focuses only on the relation between rain induced attenuation and DSD, whereas the scientifically novel results of the manuscript are rather connected to the other phenomena influencing total loss such as dew and fog, wet antenna attenuation, humidity and temperature, clutter, mixed precipitation. The part related to power-law approximation might be, in my view, even replaced by a reference on a literature, as the results in the revised paper are now mostly based on comparison between observed attenuation and attenuation derived directly from DSD.*

**RESPONSE:** We kept this, since microwave links are mainly suited to estimate rainfall. Hence, the relation between rain induced attenuation and DSD is very relevant, and is also presented based on a large DSD dataset from disdrometers. Moreover, we provide a global analysis on link rainfall retrieval (Figure 7), as well as present several case studies of rainy events.

**REVIEWER:**
*The literature review on the other phenomena is provided in the introducing paragraphs of results subsections to some extent (e.g. P11L13-21) and partly also in the introduction (wet antenna).*
*It would either make sense to rename the section to make it clear it is about DSD or preferably move the fragments of theoretical background connected to other phenomena from the result subsections to this section.*

**RESPONSE:**

We have moved the opening paragraph of section 5.3 to the theory section.

**REVIEWER:**

*P4L19-20- Please, rephrase the sentence.*

**RESPONSE:**

It is unclear to us how or why this sentence should be rephrased. The provided information is correct.

**REVIEWER:**
*P7L18-21: This part is difficult to read. Do you average other quantities than DSD derived attenuation and rainfall? Being specific would improve the clearness a lot. Anyway, isn't it at the end the same as*

*averaging the DSD over the link path with weighted mean and then deriving either path-averaged attenuation or rainfall from this path-averaged DSD?*

**RESPONSE:** Although originally it was the case that other quantities where derived, this is no longer relevant for the paper as written. We have therefore changed the wording to be more specific. Indeed, the order of aggregation does not matter, and this is not relevant for the wording of this paragraph.

**REVIEWER:**
*P8L1: Please, replace 'somewhat' with some quantitative measure (by XY %).*

**RESPONSE:**

Done.

**REVIEWER:**
*P9L4: The term 'measure of the fitness' sounds strange to me. Moreover, the rest of the sentence describes procedure rather than the measure itself. Maybe the term 'reliability' would be more appropriate here.*

**RESPONSE:**

We have changed to wording to "To assess the reliability…".

**REVIEWER:**
*P9L17-21: Please rephrase. What is exactly meant with 'it could not have been resolved from magnitude alone'?*
**RESPONSE:**

It cannot be distinguished from variability in the baseline without prior knowledge or some further processing because it falls in the same magnitude range. We have clarified this in the text.

**REVIEWER:**
*P9L40-41: Have you experienced also longer dying times as Schleiss et al., (2013) did? (I am just curious)*

**RESPONSE:**

Yes, we did. See, for example, Fig. 16 and P15L1 in the marked-up manuscript (the second revised version) where we found a signal that seems to suggest a 3 hour drying time. We added a remark in the paragraph to point this out.

**REVIEWER:**
*P1L1: Bad English, please rephrase.*

**RESPONSE:**

Although we believe this is perfectly fine English usage, we have changed the sentence to a more traditional structure.

**REVIEWER:**
*P10L14: reasonable correlation means strong correlation? Please, specify.*

**RESPONSE:**

Reasonably strong, yes. We have clarified this in the text.

**REVIEWER:**
*P10L29: What is meant with 'but drops afterwards' with respect to relation between visibility and rainfall intensity? Please, rephrase.*

**RESPONSE:**

We meant to say that the visibility drops after the rain event has already ended. This is indeed confusingly worded. Furthermore, this is actually not really relevant to the description of the event itself, so we opted to leave this out altogether.

**REVIEWER:**
*P11L13-21: Consider moving this paragraph to the Theoretical background, as this is not result.*

**RESPONSE:**

Done.

**REVIEWER:**
*P12L4: Do you mean more detailed analysis of your data, or really intent to research this phenomenon in general? The generic description of snow melting might be really challenging (and not so easy to study in Netherlands).*

**RESPONSE:**

Neither. We do not think that the data we have is sufficient to study this phenomenon and we currently have no plans or opportunity to further study this, although we do hope somebody would pick this up. The meaning of the sentence is rather to express the lack of conclusiveness. We have adapted the text to hopefully better express the intent.

**REVIEWER:**
*P12L23-24: Wrong syntax of the sentence.*

**RESPONSE:**

Again, we believe the syntax is not wrong in principle, but we have adapted it to a more traditional word order.

**REVIEWER:**

*P12-25-30:*
*The paragraph contains interesting findings but it is really uneasy to read (what is meant e.g. with 'and for the most part a linear regression makes for a good fit'?). The description of residual errors is in my view rather redundant (at least without using it further for expressing uncertainty of regression coefficient). I do not see the link to the main message of the paragraph which is at least as I interpret it i) strong linear attenuation-temperature relationship during dry weather vs. milder (or none) during dry weather ii) that this relationship is hardware specific. I think that restructuring of the paragraph is really needed.*
*Something like this might help:*
*There is a strong negative correlation between temperature and attenuation (r = -0.80 to -0.92) for the periods with relative humidity below 90%. The slope of the linear fit is substantially lower for Nokia link (-0.024 dB K-1) than for the other devices (between 0.1 and 0.2 dB K-1), although Nokia link operates at the same frequency and polarization as one of the RAL devices. The linear relationship between attenuation and temperature is milder for the periods with higher relative humidity (XXX to YYY) and completely disappears in the case of Nokia device. This indicates that the attenuation-temperature relationship is clearly more influenced by link hardware than their frequency or polarization.*

**RESPONSE:**

Agreed. We have revised the paragraph accordingly and removed redundant information.

**REVIEWER:**
*P12L40-42: Bad English, please rephrase.*

**RESPONSE:**

We believe it is not bad English, but we agree that it is quite difficult to read. We have changed the sentence structure in order to be more readable.

**REVIEWER:**
*P13L10-15: Difficult to read, please rephrase.*

**RESPONSE:**

In hindsight this paragraph indeed seems a bit rambling and difficult to follow. We have rewritten it.

**REVIEWER:**
*P13L40-42: This is unclear. How did you come to 1.5 dB and what do you mean with 'attenuation by fog droplets themselves'? Do you mean water vapor attenuation along a link path, or something else? Please clarify.*

**RESPONSE:**

We have added clarification on how we arrived at 1.5 dB. By fog droplets we mean cloud droplets (i.e. atmospheric liquid water content) rather than water vapor.

**REVIEWER:**
*P14L11: What it is exactly meant with 'antennas were wetted until saturation' and how did you recognize this saturation? Based on attenuation observations, or based on observations of antenna surface?*

**RESPONSE:**

Based on observations of the surface, i.e. visual inspection

**REVIEWER:**
*P14L20: '…decay time 3 minutes.' 3 minutes to 'half-life' value or to some other quantile of max wet antenna attenuation?*

**RESPONSE:**

To a reduction of 95 %. We have clarified this in the text.

**REVIEWER:**
*P14L22: Please quantify (provide e.g. range) the initial drop of signal level.*

**RESPONSE:**

Done.

**REVIEWER:**
*P14L26: Minda and Nakamura (2005), do not claim/conclude that water layer on the antenna surface has an uniform layer. Their model is purely empirical. Please correct. Consider also comparing your results with Schleiss et al., (2013) who reported substantially longer drying times than Minda and Nakamura (2015).*

**RESPONSE:**

We have corrected this in the revised text. We also added a few sentences relating our findings to those of Schleiss et al., (2013).

**REVIEWER:**
*P15L36: It is not clear how you do come to the conclusion that fog related attenuation is caused predominantly by antenna wetting. Please provide evidence.*

**RESPONSE:**

We admit that this is too speculative and removed this claim.

**REVIEWER:**

*P15L40: What it is meant with 'net radiation flux is away from the surface'? Please rephrase.*

**RESPONSE:**

We have rephrased this in the revised text ("net radiation flux towards the surface is negative").

**REVIEWER:**

*P16L1-15: Consider removing (or moving) the paragraph about DSD. It describes known and previously well documented findings.*

**RESPONSE:**

Although the k-R relationship we found is new, in the sense that it is based purely on data from this experiment and is subtly different from k-R relationships derived from older datasets, it is not 'novel' and more importantly not the main thrust of this paper. Therefore, we have removed it from the conclusions, as suggested by the reviewer.

**REVIEWER:**

*P16L21: '…additive and multiplicative bias seems quite consistent'. There is no quantitative inter event evaluation of additive and multiplicative bias provided in the results section. Please, provide this information in result section or reformulate this conclusion to be sufficiently supported by presented results. Furthermore, replace 'quite consistent' with a formulation indicating better (if possible quantitatively) to which extent both types of biases vary within the event.*

**RESPONSE:**

We have rephrased this sentence to better indicate what we mean.

**REVIEWER:**

*P16L38-40: This is very general recommendation which is difficult to apply especially when additional rainfall data is not available (i.e. when one could potentially benefit from microwave links the most). I believe that more specific recommendation could be provided based on the presented results, e.g. on use of links having same/similar path, etc.*

**RESPONSE:**

We do not suggest that the optimization should be based on the complete operational network. Putting devices of the same type as used in the network in a test setup or building a temporary test setup around individual links in the network representing the hardware types used should be enough to train an algorithm to hardware specific quirks.

**REVIEWER:**

*P16L40-42: Yes, however, can we recognize such link without having reference rainfall data? Could e.g. variability of baseline during dry weather period tell us something about link reliability in terms of rainfall estimation?*

**RESPONSE:**

See above. If we have test data for the different link types, we only need to recognize their names.

[revised manuscript text omitted]

---

## Author Response (AR3)

**REVIEWER**:

*The revised manuscript reflects the comments and suggestions raised in the previous stage of the review process and I, therefore, recommend it for publication. The results section is now more compact (some parts were moved to theoretical background) and presented results are described where applicable in quantitative manner. The conclusion section was restructured to follow findings presented in the result section. I only suggest one minor specific technical correction:*

*P12L5-6 (manuscript v3): 'The slopes ... are similar for all data selections (within 15 % of the slopes found for 14-20 April)'. The quantitative description of slope range in the brackets is bit unclear. I guess that 15 % describes the relative difference of the slopes to the one found for 14-20 April. It should, therefore, be something like: '(difference within +- 15 % compared to the slopes found for 14-20 April)'.*

**RESPONSE**: That is indeed the intended meaning. We agree that the suggested wording is slightly clearer and we have therefore adopted it in the manuscript.

[revised manuscript text omitted]